# Online Navigation Refinement: Achieving Lane-Level Guidance by Associating Standard-Definition and Online Perception Maps

**Jiaxu Wan[1], Xu Wang[2, ◇], Mengwei Xie[2], Xinyuan Chang[2], Xinran Liu[2], Zheng Pan[2],**

**Mu Xu[2], Hong Zhang[1, 3], Ding Yuan[1, 4], Yifan Yang[1, 4, ‡]**

{wanjiaxu, dmrzhang, dyuan, stephenyoung}@buaa.edu.cn, {wx303649, xiemengwei.xmw, changxinyuan.cxy, tom.lxr, panzheng.pan, xumu.xm}@alibaba-inc.com

[1]School of Aerospace, BUAA    [2]Amap, Alibaba Group    [3]Key Laboratory of Spacecraft Design Optimization and Dynamic Simulation Technology, Ministry of Education    [4]State Key Laboratory of High-Efficiency Reusable Aerospace Transportation Technology

Github: https://github.com/WallelWan/OMA-MAT

## Abstract

Lane-level navigation is critical for geographic information systems and navigation-based tasks, offering finer-grained guidance than road-level navigation by standard definition (SD) maps. However, it currently relies on expansive global HD maps that cannot adapt to dynamic road conditions. Recently, online perception (OP) maps have become research hotspots, providing real-time geometry as an alternative, but lack the global topology needed for navigation. To address these issues, Online Navigation Refinement (ONR), a new mission is introduced that refines SD-map-based road-level routes into accurate lane-level navigation by associating SD maps with OP maps. The map-to-map association to handle many-to-one lane-to-road mappings under two key challenges: (1) no public dataset provides lane-to-road correspondences; (2) severe misalignment from spatial fluctuations, semantic disparities, and OP map noise invalidates traditional map matching. For these challenges, We contribute: (1) Online map association dataset (OMA), the first ONR benchmark with 30K scenarios and 2.6M annotated lane vectors; (2) MAT, a transformer with path-aware attention to aligns topology despite spatial fluctuations and semantic disparities and spatial attention for integrates noisy OP features via global context; and (3) NR P-R, a metric evaluating geometric and semantic alignment. Experiments show that MAT outperforms existing methods at 34 ms latency, enabling low-cost and up-to-date lane-level navigation.

## 1 Introduction

Lane-level navigation has emerged as a critical capability in geographic information systems (GIS) Hansson et al. (2020); Guo et al. (2025) and navigation-based tasks Peng et al. (2025); Li et al. (2025); Shan et al. (2025), offering finer-grained guidance than traditional road-level navigation based on standard definition (SD) maps (Fig. 1 (a)) Zhang et al. (2024d). However, today's lane-level navigation systems mostly rely on pre-built HD maps, which are expensive to create and maintain. They often miss real-world changes Elghazaly et al. (2023)—like construction, rule updates, or accidents—leading to outdated guidance and safety risks. Recently, online perception (OP) maps have become research hotspots within autonomous driving Li et al. (2022); Liao et al. (2023b). Powered by vehicle sensors, the OP maps offer current and localized lane depictions. However, as shown in Fig. 1 (b), they inherently lack global route topology and cannot support lane-level navigation Wong et al. (2020).

To achieve up-to-date and low-cost lane-level navigation, we introduce a new mission called **Online Navigation Refinement** (ONR). The core objective of the mission is to transform the road-level

---

‡Corresponding author.    ◇Project Lead.

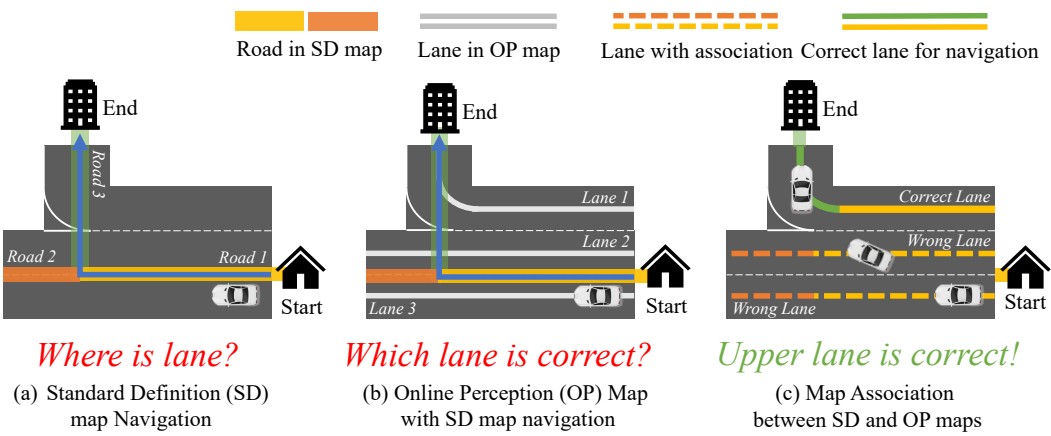

Figure 1: Motivation for online navigation refinement and map association. The roads and lanes with the same color indicate that they are interconnected: (a) Standard Definition (SD) maps offer only road-level navigation without lane details. (b) Online Perception (OP) maps offer lane-specific details, yet they are not connected to SD maps and cannot identify the correct lane. (c) Associating SD and OP maps enables lane selection, achieving online navigation refinement and lane-level navigation.

navigation derived from SD maps into precise lane-level navigation aligned with the OP maps. For a fast and accurate process, ONR needs a paradigm of map-to-map matching, called **map association**, as shown in Fig 1 (c). Compared with map matching (MM) that bind GPS trajectories to static SD/HD maps Chao et al. (2020) as a path-to-map matching based on the HMM Newson (2009), seq-to-seq model Feng et al. (2020); Ren et al. (2021), graph model Liu et al. (2023b) and transformers Tang et al. (2025), map association uses map-to-map matching because SD maps and OP maps are heterogeneous and do not share a one-to-one correspondence. Attempting to directly align SD paths with OP maps, or the other way around, overlooks the semantic disparities between the two.

Specifically, map association faces two key challenges. First, no public datasets provide structured, lane-to-road-level correspondences between SD maps and online perception (OP) maps. Although there exist auto-driving datasets (e.g., nuScenes Caesar et al. (2020a), OpenLaneV2 Wang et al. (2023)) offer local lane geometries, and OpenStreetMap provides road-level topologies, none establish explicit, learnable mappings between them. Secondly, SD and OP maps display inherent heterogeneity due to diverse granularity, resulting in substantial spatial and semantic differences. A robust association mechanism is required to manage spatial fluctuations (such as GPS drift and changes in scale) and semantic disparities (such as the differing number of lanes on a road). It should also be capable of processing noise issues in OP maps in real time, which include lane discontinuities, omissions, or errors in OP mapping. Figure 6 in Appendix D illustrates the complexity and challenge of map association.

To address these challenges, we make three core contributions: **(1) Dataset**: We introduce **Online Map Association Dataset (OMA)**, the first open source benchmark for the online navigation refinement mission in the paradigm of map association, derived from nuScenes Caesar et al. (2020a) and OpenStreetMap osm, which contains more than 30K scenarios, 480K road paths and 2.6M lane vectors with manually annotated associations. **(2) Baseline**: We present **MAT**, a lightweight transformer-based model for the real-time association of maps. To handle spatial and semantic differences, MAT incorporates two key modules: path-aware attention and spatial attention. Path-aware attention reorders and group vector tokens by path index, implicitly encoding topological structure to align SD and OP maps despite spatial offsets. Spatial attention sorts and groups road/centerline tokens using a spatial curve, enabling global context modeling through cross-topological feature integration. **(3) Metric**: We propose **Navigation Refinement P-R (NR P-R)**, a metric that measures alignment of the path and the correspondence through precision, recall and F1 specifically designed for map association. Traditional MM metrics only measure the accuracy of the matching, but the NR P-R evaluates both the geometric similarity and the accuracy of the matching. Moreover, NR P-R only utilizes annotation in ground-truth perception maps, rendering the benchmark suitable for the evaluation of any map generation method.

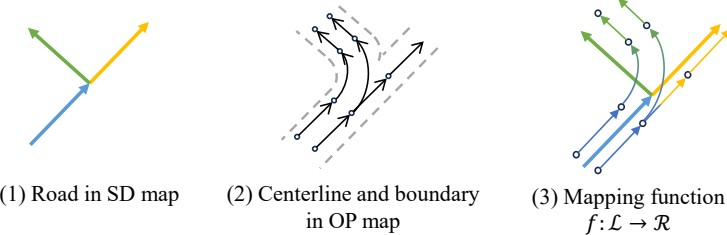

(1) Road in SD map      (2) Centerline and boundary in OP map      (3) Mapping function $f : \mathcal{L} \to \mathcal{R}$

Figure 2: (a) The schema of SD map input: Road $\mathcal{R}$. (b) The schema of OP map input: Centerline $\mathcal{L}$ and boundary $B$. (c) The objective in our task: Mapping function $f$.

Extensive experiments indicate that MAT achieves state-of-the-art performance with 34ms latency on OMA, showing significant improvements compared to traditional map matching methods and deep learning map matching method, enabling low-cost and up-to-date lane-level navigation.

## 2 RELATED WORK

**Map Matching.** Map matching investigates how to link GPS tracks to SD or HD maps. Hansson et al. (2020) utilizes a hidden Markov model (HMM) considering trajectories as observations and the corresponding road segments as states, demonstrating superiority over earlier geometric matching algorithms. In the realm of deep learning, the main approach involves sequence-to-sequence (Seq2Seq) methods Feng et al. (2020); Ren et al. (2021). With an input trajectory consisting of a sequence of geo-points, the encoder-decoder framework provides a sequence of matched road segments. Recently, GraphMM Liu et al. (2023b) has used a graph-based approach that explicitly integrates all the correlations mentioned above, while EAM[3] Tang et al. (2025) uses a BERT-like transformer Alaparthi & Mishra (2020) to achieve the best performance. Compared to the map matching method, MAT uses a map-to-map paradigm to accommodate the many-to-one correspondence between lanes and roads in real time. Moreover, unlike the self-attention used in EAM[3], MAT employs spatial and path-aware attention by incorporating the spatial curve and the path index, thus striking a balance between performance and latency.

**Map Generation and Priors.** The construction of online perception maps is a popular topic in autonomous driving Hao et al. (2024); Zhang et al. (2024a); Li et al. (2024) and is crucial for subsequent tasks Wan et al. (2025); Zhang et al. (2024b). HDMapNet Li et al. (2022) pioneered BEV-based map generation through sensor fusion, while LSS Philion & Fidler (2020) introduced depth-aware BEV transformation. VectorMapNet Liu et al. (2023a) enabled end-to-end vector prediction, and the MapTR Liao et al. (2023a) series introduced hierarchical query embeddings for instance-level construction. We adapt MapTRv2 Liao et al. (2023b) for online HD construction using the annotations from our dataset and demonstrate its compatibility with the SD-HD association. Recent works such as SMERF Luo et al. (2024), P-MapNet Jiang et al. (2024), and TopoSD Yang et al. (2024) leverage SD map priors to mitigate sensor noise and refine perception geometry. Unlike these approaches that focus on enhancing generation quality, MAT targets the explicit *association* phase, utilizing topological optimization to resolve assignment ambiguity arising from residual noise.

## 3 TASK DEFINITION

The goal of online navigation refinement is to transform road-topology-aligned navigation into lane-topology-aligned navigation. To achieve this, we utilize the map association as a map-to-map match paradigm. This involves correlating SD maps with online perception maps, allowing us to translate any route from an SD map into the corresponding route on an online perception map.

**Standard Definition Map (SD Map).** As shown in Fig.2 (a), SD maps represent road networks that use roads as primary primitives. Formally, a SD map is defined as a graph $\mathcal{G}_{\mathcal{R}} = (\mathcal{R}, \mathcal{E}_{\mathcal{R}})$, where $\mathcal{R} = \{r_1, r_2, \ldots, r_m\}$ denotes a set of roads and $\mathcal{E}_{\mathcal{R}}$ encodes their topological connectivity. Each road $r_j$ is parameterized by a sequence of directed vectors:

$$r_j = (\overrightarrow{q_{j1}q_{j2}}, \overrightarrow{q_{j2}q_{j3}}, \ldots, \overrightarrow{q_{jk-1}q_{jk}}), \quad q_{jk} \in \mathbb{R}^2, \tag{1}$$

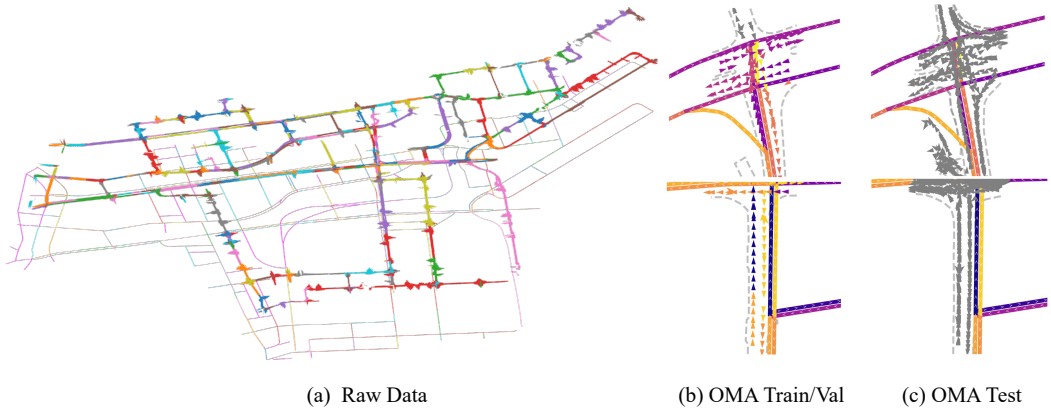

(a) Raw Data          (b) OMA Train/Val     (c) OMA Test

Figure 3: (a) The visualization of SD map and GT OP map with association annotations of Boston in nuScenes. The same color implies an associative pair. (b) The visualization of OMA Train/Val set. (c) The visualization of OMA Test set.

where consecutive points define road segments through uniform spatial sampling.

**Online Perception Map (OP Map).** As shown in Fig.2 (b), OP maps provide details of the lane level, primarily represented as a center line network. We model an OP map as a graph $\mathcal{G}_{\mathcal{L}} = (\mathcal{L}, \mathcal{E}_{\mathcal{L}})$, where $\mathcal{L} = \{l_1, l_2, \ldots, l_n\}$ is a set of centerlines, each sampled at uniform intervals:

$$l_i = \overrightarrow{p_i^1 p_i^2}, \quad p_i^1, p_i^2 \in \mathbb{R}^2. \tag{2}$$

$\mathcal{E}_{\mathcal{L}}$ captures topological relations between the adjacent centerlines. In addition, we include road boundary vectors $\mathcal{B} = \{b_1, b_2, \ldots, b_{m_b}\}$, which reflect the extent and shape of the actual road. Each boundary $b_j$ is specified as:

$$b_j = \left( \overrightarrow{h_{j1} h_{j2}}, \overrightarrow{h_{j2} h_{j3}}, \ldots, \overrightarrow{h_{jk-1} h_{jk}} \right), \quad h_{jk} \in \mathbb{R}^2. \tag{3}$$

**Objective**. As shown in Fig.2 (c), given $\mathcal{G}_{\mathcal{R}}$ and $\mathcal{G}_{\mathcal{L}}$, the task is to learn a mapping function $f : \mathcal{L} \to \mathcal{R}$ that assigns each centerline $l \in \mathcal{L}$ to its corresponding road $r_l \in \mathcal{R}$. The function satisfies two key constraints: 1. *Uniqueness*: Each centerline $l$ corresponds to exactly one ground truth road $r_l$; 2. *Multiplicity*: A single road $r \in \mathcal{R}$ may be associated with multiple centerlines $l_1, l_2, \ldots \in \mathcal{L}$.

This formulation casts the alignment task as a multi-to-one classification problem, where the number of classes equals the number of $|\mathcal{R}|$, and each centerline acts as an input sample. The goal is to maximize classification accuracy while preserving topological consistency between SD and OP maps. After we form this association, any path on the SD map allows us to locate all matching paths on the OP map through a single topological sorting.

## 4 DATASET AND METRIC

### 4.1 DATASET OVERVIEW

The source of ground-truth OP maps is nuScenes Caesar et al. (2020b), which includes locations in Boston and Singapore, featuring centerline geometries scanned with LiDAR. The SD maps were obtained from OpenStreetMap (OSM). We manually annotated the data to establish the association between the ground truth of the OP map and the SD map. The visualization is shown in Fig. 3 (a), which shows the global SD and ground-truth OP map of Boston with an association annotation.

The dataset is divided into training, validation, and test sets (see Fig. 3 (b) and (c)). Within the nuScenes framework, we employ the pon-split of nuScene. This approach designates distinct areas for training and validation/testing datasets, effectively preventing data leakage. The validation and test data sets are generated from the same scenes, but they have a variety of OP maps. For the OP map, both training and validation use ground-truth maps with manual correspondence annotations. For testing, the OP map is generated by MapTRv2 and its annotation is the same as the scene in the val set. It is essential to explain that the validation set and test set in OMA have equal significance.

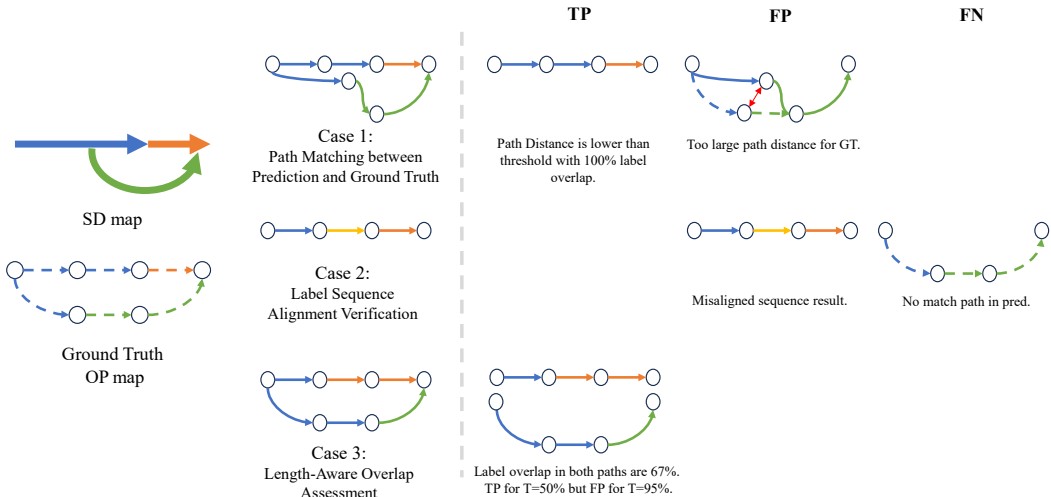

Figure 4: Example of TP, FP and FN for evaluate Navigation Refinement Precision-Recall.

The purpose of the validation set is to assess the association capability of the OP map under noise-free conditions, whereas the test set evaluates this capability when the OP map experiences significant noise. This distinction sets apart the validation/test split in OMA from that in typical datasets. The specific analysis is provided in the Appendix D.

## 4.2 EVALUATION METRIC

Standard map matching metrics, such as Trajectory-Level Accuracy, assume one-to-one mappings and focus on path sequencing. In contrast, map association requires evaluating noisy OP maps against ground truth. We propose the Navigation Refinement pr metric—adapted from Reachability pr Lu et al. (2023)—to assess map generation performance based exclusively on ground-truth annotations.

**Navigation Refinement P-R**: Fig. 4 provides a detailed description of typical scenarios encountered during the Navigation Refinement Precision Recall (P-R) evaluation. The examination protocol involves three consecutive stages:

*1. Path Matching:* Predicted paths are matched to GT via a 1m bidirectional Chamfer distance threshold Lu et al. (2023). This step is vital as only GT paths contain association annotations. Non-matches are labeled FP and excluded from SD-HD analysis. As in Fig. 4 (Row 1), low-deviation paths align (left), while high-deviation ones yield FPs (middle).

*2. Label Sequence Alignment:* Successful matches are converted into SD map link sequences. Alignment failures are marked FP/FN. Case 2 in Fig. 4 shows sequence misalignment leading to an FP (middle) and an unmatched GT as an FN (right).

*3. Length-Aware Overlap:* Semantic consistency is measured by a Label Overlap score—the length ratio of road-ID-consistent segments to the total path. TP classification depends on threshold $T$. As shown in Fig. 4 (Case 3), 67% overlap is a TP at $T = 50\%$ but an FP at $T = 95\%$.

Following the mAP conventions Lin et al. (2014), we use $T = [0.5 : 0.05 : 0.95]$ (10 thresholds) and report mean P-R and F1 scores. To mitigate path-length bias, we separately compute metrics across 15 length intervals $L = [[0, 5), [5, 10), .., [70, +\infty)]$ before aggregation. The ablation study of distance threshold and length intervals is shown in the appendix G.

## 5 METHOD

### 5.1 OVERALL ARCHITECTURE

As depicted in Fig. 5, the Map Association Transformer (MAT) is a transformer specifically designed for real-time map association. All inputs are vectorized representations $\mathcal{V} = \{\vec{v}_1, \vec{v}_2, \ldots, \vec{v}_N\}$, where each vector $\vec{v}_i$ is parameterized by two endpoints and direction: $\vec{v}_i = [p_{i1}^x, p_{i1}^y, p_{i2}^x, p_{i2}^y, \theta_i]$, with $\theta_i = \arctan\left(\frac{p_{i2}^x - p_{i1}^x}{p_{i2}^y - p_{i1}^y}\right)$ and $p_{i1}, p_{i2} \in \mathbb{R}^2$ being the start/end points. The input maps are

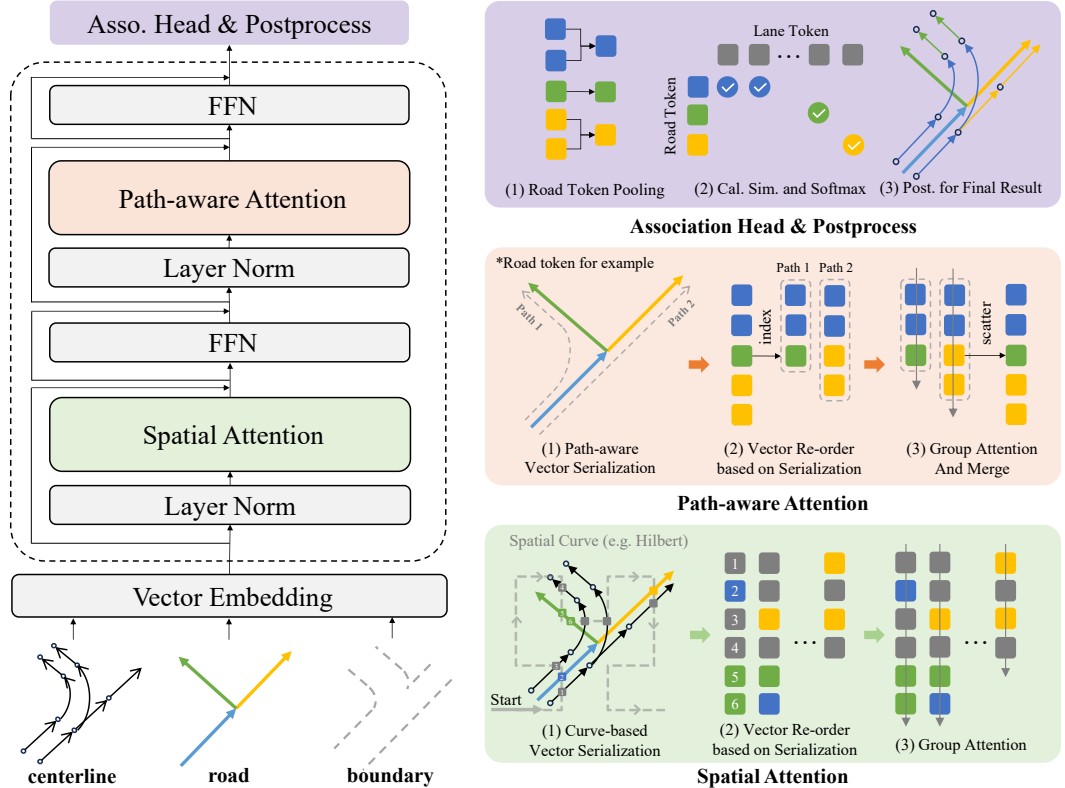

Figure 5: **Overview of Map Association Transformer (MAT).** The framework processes vectorized roads in SD map and centerlines/boundaries in OP map through $N$ stacked layers containing Spatial Attention (for global context via curve-based serialization) and Path-Aware Attention (for topological alignment via path indexing). The Association Head then aggregates road features and calculates association probabilities with centerline tokens to generate the final navigation refinement result.

composed of an SD map, an OP map, and a boundary. The SD map ($\mathcal{G}_\mathcal{R}$) comprises road vectors $\mathcal{R} = \{r_1, \ldots, r_{m_r}\}$, which form a graph with topological edges $\mathcal{E}_\mathcal{R}$. Each road $r_j$ is transformed into an ordered sequence of vectors $\mathcal{R}_j = \overrightarrow{q_{j1}q_{j2}}, \overrightarrow{q_{j2}q_{j3}}, \ldots$ through its parameterized segments. The OP map ($\mathcal{G}_\mathcal{L}$) consists of centerline vectors $\mathcal{L} = \{l_1, \ldots, l_{m_l}\}$ that represent the centerlines. Each centerline $l_i$ is transformed into vectors $\mathcal{L}_i = \overrightarrow{p_i^1 p_i^2}, \overrightarrow{p_i^2 p_i^3}, \ldots$ based on consecutive points $p_i^j$. The boundary ($\mathcal{B}$) includes the boundary vectors $\mathcal{B} = \{b_1, \ldots, b_{m_b}\}$, converted similarly to the roads: $b_j \to \mathcal{B}_j = \overrightarrow{h_{j1}, h_{j2}}, \overrightarrow{h_{j2}h_{j3}}, \ldots$. These vectors are processed by the vector embedding module, which maps each 5D vector $\vec{v}_i$ to a high-dimensional feature $F_{\vec{v}_i} \in \mathbb{R}^C$ via a two-layer MLP. The outputs are aggregated into feature matrices: $F_{road} \in \mathbb{R}^{N_r \times C}$, $F_{centerline} \in \mathbb{R}^{N_l \times C}$, and $F_{boundary} \in \mathbb{R}^{N_b \times C}$, where $N_{(\cdot)}$ denotes the total number of vectors (e.g. $N_r = \sum_{j=1}^{m_r} \text{length}(\mathcal{R}_j)$).

Subsequently, the vector tokens are input into a transformer network. MAT consists of stacked MAT blocks to extract hierarchical features, each block containing Path-Aware Attention (PA), Spatial Attention (SA), and feed-forward network (FFN). In the association head, each road token requires the pooling to obtain a representative token $\bar{F}_{road}$ for the current road. The association of the centerline with the road is calculated by combining the attention between $\bar{F}_{road}$ and $F_{centerline}$, generating the probability distribution of the associations of the SD-OP map. The association probabilities are further refined by a post-processing method to enforce topological constraints. The details of the implementation and the module are provided in Appendix F.

## 5.2 PATH-AWARE ATTENTION

Path-Aware Attention (PA) is designed to extract locally stable geometric features that preserve the structural arrangement of elements.

**Topological Ordering for Efficient Attention.** To achieve real-time inference, our framework employs Group Attention, which reduces computational complexity from $O(N^2)$ to linear $O(N)$ by restricting interactions to local windows. However, this efficiency comes with a constraint: the effectiveness of group attention depends heavily on the semantic meaningfulness of token order. PA addresses this by introducing **Topological Ordering** as an inductive bias. Unlike random ordering, which would fragment the graph into unrelated segments, PA explicitly models long-range dependencies by constructing paths from root to leaf nodes. This ensures that topologically connected predecessor and successor nodes are placed adjacently in the sequence.

**Mechanism.** Specifically, we identify all valid paths from a starting point to an endpoint. We then reorder vector tokens to align with these path indices. This strict ordering allows the subsequent grouped attention (with size $k$) to focus solely on topologically relevant neighbors. Finally, tokens are reversed to their original sequence, and features of tokens appearing in multiple paths are averaged.

### 5.3 SPATIAL ATTENTION

While PA captures topological connectivity, it may miss interactions between geometrically adjacent but topologically distant segments (e.g., parallel lanes or disconnected road boundaries). The Spatial Attention (SA) mechanism complements PA by capturing instance-level interactions across a wider spatial scope via **Spatial Ordering**.

**Motivation: Vector Serialization as Geometric Clustering.** Similar to PA, SA relies on a specific ordering strategy to maximize the efficacy of group attention. However, instead of following graph connectivity, SA utilizes **Vector Serialization** to cluster tokens based on geometric proximity. This serves as a spatial inductive bias: it forces physically adjacent entities—even if they belong to different map layers or are disconnected—to be placed adjacently in the 1D sequence. Consequently, when the sequence is sliced into groups, highly correlated tokens naturally fall into the same attention bucket, enabling the model to handle GPS offsets and map alignment robustly.

**Mechanism: Attention with Vector Serialization.** As illustrated in Fig. 5 green part, we implement this spatial ordering through the following steps:

*1) Coordinate Discretization:* Each vector token $\vec{v}_i$ is encoded into a 3D discrete coordinate $(x, y, r)$, representing the quantized grid location and orientation of the vector.

*2) Serialization via Space-Filling Curves:* We employ a space-filling curve function $\varphi^{-1}$ (e.g., the Hilbert curve Hilbert (1935)) to map the 3D coordinates to a single 1D index. This mapping is crucial as it preserves spatial locality in the 1D domain better than simple row-major scanning.

*3) Grouped Attention & Restoration:* Tokens are reordered based on their 1D indices. Self-attention is performed within each group to aggregate spatial context, followed by an inverse operation to restore the original order.

### 5.4 ASSOCIATION AND LOSS FUNCTION

**Association.** The association between the roads and the centerline is calculated through a mechanism of cross-attention. For each road $j$, we first aggregate its token features $\{F_{j1}^{road}, \ldots, F_{jN}^{road}\}$ into a representative feature $\bar{F}_j^{road} = \frac{1}{N} \sum_{n=1}^{N} F_{jn}^{road}$, where $N$ denotes the number of road tokens on roads $r_j$ and $F_{jn}^{road} \in \mathbb{R}^d$. The association probability $Prob_{ij}$ between the centerline $i$ and the road $j$ is then calculated as:

$$Prob_{ij} = \exp\left(\frac{F_i^{cl} \cdot \bar{F}_j^{road}}{\sqrt{d}}\right) / \sum_{k=1}^{K} \exp\left(\frac{F_i^{cl} \cdot \bar{F}_k^{road}}{\sqrt{d}}\right), \tag{4}$$

where $F_i^{cl} \in \mathbb{R}^d$ is the token feature of the centerline, $d$ is the dimension of the feature, and $K$ represents the total number of roads. This formulation normalizes the similarity scores in all roads for each centerline $i$, ensuring a valid probability distribution.

**Loss Function.** Followed by Liu et al. (2023b); Ren et al. (2021), we optimize the model using a combination of cross-entropy loss (CE) and connection temporal classification (CTC) loss. The total loss is a weighted sum:

$$\mathcal{L}_{\text{total}} = \alpha \cdot \mathcal{L}_{\text{CE}} + \beta \cdot \mathcal{L}_{\text{CTC}}, \tag{5}$$

with hyperparameters $\alpha$ and $\beta$ balancing the two objectives. In practice, $\alpha = 1, \beta = 0.01$.

Table 1: Result on OMA Val set. La. means latency. MM, PM, GM, MA means map matching, graph matching, point matching and map association method. P-M and M-M means the model is trained and inferred by path-to-map or map-to-map.

| Methods | Present | Type | Paradigm | Val set | |
|---|---|---|---|---|---|
| | | | | NR-F1$^{50:95}$ | La./ms |
| HMM Newson (2009) | SIGSPATIAL'09 | MM | P-M | 70.1 | 465 |
| DeepMM Feng et al. (2020) | TMC'20 | MM | P-M | 70.6 | 328 |
| MTrajRec Ren et al. (2021) | KDD'21 | MM | P-M | 71.2 | 849 |
| GraphMM Liu et al. (2023b) | TKDE'23 | MM | P-M | 72.3 | 469 |
| EAM$^3$ Tang et al. (2025) | TITS'25 | MM | P-M | 72.9 | 345 |
| KNN Cochran (1965) | – | GM | M-M | 68.6 | 299 |
| GMT He et al. (2024) | TPAMI'24 | GM | M-M | 68.8 | 93 |
| FastMAC Zhang et al. (2024e) | CVPR'24 | PM | M-M | 70.3 | 62 |
| MAT-T (Ours) | – | MA | M-M | 78.2 | **34** |
| MAT-L (Ours) | – | MA | M-M | **78.7** | 70 |

Table 2: Result on OMA Test set. La. means latency. MM, PM, GM, MA means map matching, graph matching, point matching and map association method. P-M and M-M means the model is trained and inferred by path-to-map or map-to-map. SGG. means SeqGrowGrpah Xie et al. (2025a).

| Methods | Present | Type | Paradigm | NR-F1$^{50:95}$ on Test Sets | | | La./ms |
|---|---|---|---|---|---|---|---|
| | | | | MapTR | MapTRv2 | SGG. | |
| HMM Newson (2009) | SIGSPATIAL'09 | MM | P-M | 33.1 | 36.0 | 46.5 | 561 |
| DeepMM Feng et al. (2020) | TMC'20 | MM | P-M | 32.8 | 33.5 | 43.9 | 733 |
| MTrajRec Ren et al. (2021) | KDD'21 | MM | P-M | 34.1 | 36.8 | 48.9 | 1593 |
| GraphMM Liu et al. (2023b) | TKDE'23 | MM | P-M | 35.9 | 38.7 | 49.2 | 889 |
| EAM$^3$ Tang et al. (2025) | TITS'25 | MM | P-M | 36.3 | 39.1 | 50.8 | 679 |
| KNN Cochran (1965) | – | GM | M-M | 31.7 | 34.6 | 43.5 | 313 |
| GMT He et al. (2024) | TPAMI'24 | GM | M-M | 30.1 | 33.4 | 42.3 | 105 |
| FastMAC Zhang et al. (2024e) | CVPR'24 | PM | M-M | 32.0 | 35.8 | 45.0 | 81 |
| MAT-T (Ours) | – | MA | M-M | 41.5 | 44.8 | 54.8 | **35** |
| MAT-L (Ours) | – | MA | M-M | **41.9** | **45.0** | **54.9** | 74 |

## 5.5 TOPOLOGY POST-PROCESS

We formalize topological decoding as a structured prediction on the entire path of the centerline $\mathcal{P}_j$, $j \in [1, \cdots, K]$. $K$ is the number of total paths. The two-stage decoding process operates as follows:

**Token Initialization**. For each centerline path $\mathcal{P}_j$, we select the initial centerline $T_{\max}$ via:

$$T_{\max} = \underset{l \in \mathcal{P}_j}{\operatorname{argmax}} \max_{r \in \mathcal{R}} P(l, r) \tag{6}$$

where $P(l, r)$ is the probability of association from centerline $l$ to road $r$.

**Topological-constraint Beam Search**. Based on beam search, topological constraint beam search makes the following two improvements: 1. Modify the one-way search to implement a bidirectional search starting at $T_{max}$. 2. When generating new predictions, instead of using the approach of taking the maximum value from all roads, we decode under the constraint of connectivity provided in the road network $\mathcal{E}_r$, thus ensuring that the connectivity of the road sequence corresponding to the lane path in the decoding result is consistent with the representation of the road network. Detailed expressions of the topological constraints beam search, including formula descriptions, are included in the appendix F.

Table 3: Ablation study of structure. Post. means post process. Bd. means Boundary. La. means latency.

| SA | PA | Bd. | Post. | NR-F1$^{50:95}$ | La./ms |
|----|----|-----|-------|-----------------|--------|
| | Baseline (PTv3) | | | 61.8 | **59** |
| ✓ | | | | 62.1 | 77 |
| | | ✓ | | 74.1 | 61 |
| ✓ | ✓ | | | 77.8 | 64 |
| ✓ | ✓ | ✓ | | 78.5 | 69 |
| ✓ | ✓ | ✓ | ✓ | **78.7** | 70 |

Table 4: Ablation study of loss and pool method. La. means latency.

| CE | CTC | Avg. | Max | NR-F1$^{50:95}$ | La./ms |
|----|-----|------|-----|-----------------|--------|
| ✓ | | ✓ | | 78.4 | 70 |
| | ✓ | ✓ | | 67.7 | 70 |
| ✓ | ✓ | | ✓ | 78.5 | 70 |
| ✓ | ✓ | ✓ | | 78.7 | 70 |

# 6 EXPERIMENT

## 6.1 IMPLEMENTATION DETAILS

We train our models from scratch for a total of 50 epochs using the AdamW optimizer. A cosine-decay learning rate scheduler is employed, incorporating a linear warm-up phase of two epochs. The initial learning rate, weight decay, and batch size are set to 0.0001, 0.05, and 128, respectively. All experiments are conducted on NVIDIA A6000 GPUs. The latency is measured on an NVIDIA A6000 GPU paired with an Intel(R) Xeon(R) Platinum 8369B CPU. Note that MAT-T and MAT-L share identical architectural components and training configurations, differing only in the number of Transformer blocks to balance real-time efficiency and model capacity, which details are shown in Tab. 22 in Appendix.

## 6.2 RESULT

To evaluate the performance of different methods on the OMA dataset, we categorize existing approaches into map matching methods Newson (2009); Feng et al. (2020); Ren et al. (2021); Liu et al. (2023b); Tang et al. (2025), graph matching methods He et al. (2024), and point matching methods Zhang et al. (2024e). For map matching, we traverse every path on the OP map as a GPS trajectory to match against the SD map. For graph and point matching, we treat the SD and OP maps as separate graphs or point clouds.

**Val set.** As shown in Table 1, MAT achieves the optimal balance between precision and efficiency on the validation set. Specifically, compared to traditional, Seq2Seq, and graph-based map matching methods, MAT-T demonstrates significant improvements in NR-F1$^{50:95}$, outperforming HMM Newson (2009) by 8.1%, DeepMM Feng et al. (2020) by 7.6%, MTrajRec Ren et al. (2021) by 7.0%, and the recent EAM[3] Tang et al. (2025) by 5.3%, all while maintaining a low inference latency of 34 ms. Moreover, MAT-T achieves superior performance compared to graph matching techniques like GMT He et al. (2024) and point matching approaches such as FastMAC Zhang et al. (2024e), proving its effectiveness in handling the heterogeneity between SD and OP maps which typically challenges traditional matching or point cloud algorithms.

**Test set.** To further verify the model's robustness against varying noise patterns crucial for real-world deployment, we extended the evaluation on the Test set to include OP maps generated by three distinct methods: MapTR Liao et al., MapTRv2 Liao et al. (2023b), and SeqGrowGraph Xie et al. (2025a), as summarized in Table 2. The results demonstrate that our proposed MAT method consistently outperforms all state-of-the-art baselines across these disparate generators without requiring specific fine-tuning. On the SeqGrowGraph part, MAT-L achieves an NR-F1$^{50:95}$ of 54.9%, surpassing the strongest baseline EAM[3] Tang et al. (2025) by 4.1%, while on the more challenging MapTR dataset, which exhibits different topological error patterns, MAT-L maintains a significant lead of 5.6% over EAM[3]. This consistent superiority confirms that MAT's architecture effectively generalizes across distinct noise distributions—ranging from fragmentation in MapTR to connectivity issues in growing-based methods—thereby validating the strong cross-generator generalization ability of the proposed Online Navigation Refinement framework.

## 6.3 ABLATION STUDY

The ablation study experiment is conducted with MAT-L in the val set of OMA, with latency measured using a NVIDIA A6000.

Table 5: Cross-validation between Boston and Singapore in the OMA dataset

| City | Val | Test |
|------|-----|------|
| Singapore $\rightarrow$ Singapore | 77.5 | 49.5 |
| Boston $\rightarrow$ Singapore | 77.3 | 48.7 |
| Boston $\rightarrow$ Boston | 79.4 | 42.9 |
| Singapore $\rightarrow$ Boston | 79.0 | 42.6 |

Table 6: Data Efficiency experiment on OMA dataset.

| Ratio | 1% | 2% | 5% | 10% | 20% | 50% | 100% |
|-------|----|----|----|-----|-----|-----|------|
| Val / NR-F1$^{50:95}$ | 24.9 | 64.4 | 77.1 | 77.7 | 77.3 | 78.3 | 78.7 |
| Test / NR-F1$^{50:95}$ | 22.3 | 41.5 | 44.5 | 44.6 | 46.6 | 44.7 | 45.0 |

**Structure.** As shown in Tab. 3, ablating path-aware (PA) and spatial attention (SA) reveals that PA+SA achieves the highest NR-F1$^{50:95}$ (+3.7% vs. PA-only, +15.7% vs. SA-only). PA-only outperforms SA-only (74.1% vs. 62.1%), confirming the critical role of topological awareness. Boundary improve +0.7% with a 5ms latency cost. Post-processing further enhances accuracy (+0.2%) without efficiency trade-offs.

**Loss Function and Road Pooling Method.** Tab. 4 shows an ablation study on loss functions and road pooling methods. The model exhibits strong robustness, with consistent performance across different settings, indicating low sensitivity to these components. Combining CE and CTC losses improves NR-F1 by +0.3% over CE alone and by +11.0% over CTC alone, while all variants perform competitively. Average pooling outperforms maximum pooling by only +0.2%, further confirming the model's insensitivity to pooling strategy. These results suggest that the effectiveness of the method is due to its inherently robust design, rather than specific loss or grouping choices.

**Cross-validation.** To evaluate the generalization of the model in geographically and behaviorally distinct driving environments, we conducted cross-validation in Tab. 5 using data from Boston and Singapore, two cities that differ markedly in road layout, traffic density, and regulations. The model maintains robust performance without fine-tuning: in the val set, the Boston-Singapore transfer incurred only a 0.4-point drop (79.4→79.0) and Singapore-Boston a 0.2-point drop (77.5→77.3); on the test set, the drops were similarly minor: 0.8 points (49.5→48.7) and 0.3 points (42.9→42.6), respectively.

**Data Efficiency.** To investigate the label efficiency of MAT and validate its potential for semi-supervised learning, we conducted experiments using varying subsets of the training data (ranging from 1% to 100%) on MAT-L. The results, summarized in Tab. 6, demonstrate exceptional data efficiency. In particular, with only **5%** of the annotated training data, the model achieves a $NR - F1^{50:95}$ of 77.1% in the validation set and 44.5% in the test set. These results are comparable to the performance achieved using 100% of the data (78.7% / 45.0%). This saturation in a low-data regime indicates that MAT effectively captures robust topological features without requiring massive amounts of dense annotations.

**More Ablation Study, Visualization and Failed cases.** Appendix G has shown more study of ablation study of hyperparameters of PA and SA, input size of SD map, beam width of post-process, visualizations, and failed cases.

## 7 CONCLUSION

We propose Online Navigation Refinement (ONR), a new mission that fuses static SD maps with real-time perception for accurate, low-cost lane-level navigation. To achieve ONR, we make three core contributions: We release OMA, the first public benchmark for map associations, and MAT, a transformer-based model for aligning noisy maps. We also introduce Navigation Refinement P-R, a metric evaluating both geometry and association accuracy. For limitations, we will explore end-to-end integration with motion planners, cross-domain adaptation, and unsupervised association learning to reduce annotation dependence.

## ACKNOWLEDGMENTS

This work was supported by the National Natural Science Foundation of China (62433003, 62476017).

## ETHICS STATEMENT

This work uses only publicly available autonomous driving datasets (e.g., nuScenes) and Open-StreetMap (OSM), following their licenses and usage terms. We rely solely on map-level and sensor-derived geometric information and do not use or release any personally identifiable data such as faces, license plates, or raw trajectories linked to individuals. All additional annotations were produced by trained annotators on de-identified map representations. Our method and dataset are intended for research purposes to improve road safety and navigation; any real-world deployment should include thorough testing, safety checks, and compliance with relevant regulations to avoid potential harms from incorrect or misleading guidance.

## REPRODUCIBILITY STATEMENT

This work is based entirely on publicly available datasets, specifically the nuScenes autonomous driving dataset and OpenStreetMap (OSM). We describe our model architectures, training objectives, and optimization hyperparameters, as well as dataset preprocessing, train/validation/test splits, and evaluation metrics in the main text and appendix to enable independent verification. Upon publication, we will release our code, including scripts for data preprocessing, map association construction, model training, and evaluation, together with configuration files specifying all hyperparameters, random seeds, and implementation details (e.g., framework and library versions). We will also release the processed map-association annotations introduced in this paper, together with the code of our models and trained checkpoints.

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

CONTENTS OF APPENDIX

## A    LLM USAGE

The paper has employed LLM tools such as GPT and Writefull for enhancing and refining writing in the abstract, main text, and appendix.

## B    QUESTION AND ANSWER

We have included some Q&A to enable readers to grasp the information in our dataset. Every response follows the double-blind principle.

### B.1    MOTIVATION

**For what purpose was the dataset created?**  We introduce this dataset to refine online navigation by transforming road-level paths on SD maps into lane-level paths on online perception maps. This approach facilitates cost-effective and highly real-time lane-level navigation. Current lane-level navigation systems are highly dependent on expensive and globally updated HD maps, which are costly and lag in real-time efficiency. Online perception maps, a significant focus in autonomous driving research, create local HD maps near vehicles using real-time onboard sensor data. However, they lack global topology connections, making them unsuitable for navigation. This dataset envisions aligning SD maps with online perception maps through a map-to-map association paradigm, translating road-level paths on SD maps into lane-level paths on online perception maps.

**What is the relationship between this paper and planning methods?** The task and methodology of this paper focus on delivering road-level navigation on vehicles and drivers' devices. Addresses the issue of road-level navigation provision in a context where HD maps are no longer used. Recently, research like NavigScene Peng et al. (2025) has incorporated SD navigation data into autonomous driving planning modules. We suggest that our approach can act as a preliminary stage for NavigScene, enhancing the SD navigation data to offer lane-level guidance. This grants NavigScene's planning module more detailed navigation details, thereby increasing planning precision.

### B.2    DISTRIBUTION

**Will the dataset be distributed to third parties outside the entity (e.g., company, institution, organization) on behalf of which the dataset was created?**  Yes, the dataset can be accessed publicly on the Internet.

**How will the dataset be distributed (for example, tarball on website, API, GitHub)?** The dataset will be released on GitHub and Huggingface.

### B.3    MAINTENANCE

**Who will be supporting/hosting/maintaining the dataset?** The authors will be supporting, hosting, and maintaining the dataset.

**How can the owner / curator / manager of the dataset be contacted (e.g., email address)?** You can contact the author by email which will be included in the accepted version of the paper.

**Is there an erratum?** No. We will make a statement if there is any.

**Will the dataset be updated (e.g., to correct labeling errors, add new instances, delete instances)?** Yes. we intend to refresh OMA's on-line HD map with more robust baselines in the future. The new method will be selected based on comprehensive criteria that include performance measures such as accuracy, efficiency, and generalization.

**Will older versions of the dataset continue to be supported/hosted/maintained?** Yes.

**If others want to extend/augment/build on/contribute to the dataset, is there a mechanism for them to do so?**  Yes. We will make a guide document on the website of OMA.

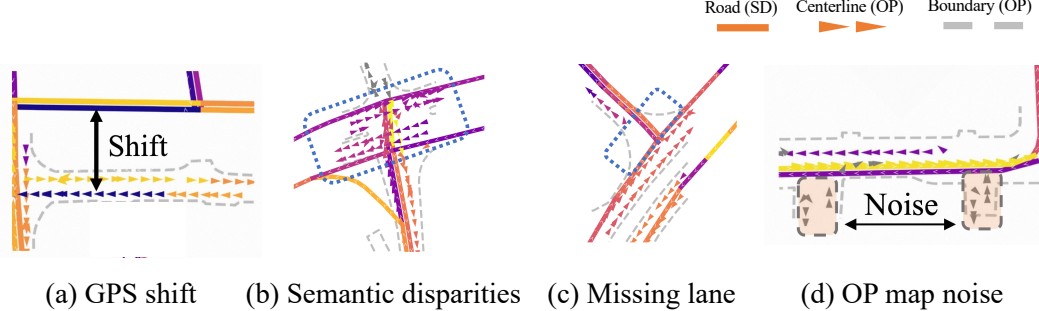

(a) GPS shift      (b) Semantic disparities      (c) Missing lane      (d) OP map noise

Figure 6: Challenge for map association. (a) GPS shift, (b) Semantic disparities, (c) missing lane, (d) noise in OP map

### B.4 COMPOSITION

**What do the instances that comprise the dataset represent?** In OMA, the basic element is a vector that encompasses the road vectors, the lane vectors, and the boundary vectors. Each vector is defined by two points that indicate its position on the map. We have normalized each vector in OMA to match the ego perspective used in real-world perception settings. Furthermore, each vector is equipped with a set of connectivity relationships that specify the other vectors to which it is connected within the road network. Road vectors include an additional road ID to identify their associated road. Lane vectors have been manually marked to show their association with a specific road.

**How many instances are there in total (of each type, if appropriate)?** OMA contain over 30K scenarios, 480K road path and 2.6M lane vector with manually annotated associations.

**Are relationships between individual instances made explicit?** On the SD map, all roads, and on the GT OP map, all lanes within the training and validation sets have thorough descriptions and associated annotations. For the test set, we offer descriptions of roads and lanes, but without associated annotations. Nevertheless, the test set can still be evaluated using annotations from the ground truth OP map in the validation set, thanks to the NR P-R metrics introduced in our paper.

**Are there recommended data splits (for example, training, development / validation, testing)?** Yes. We have already partitioned our dataset into three distinct splits: training, validation, and testing.

### B.5 COLLECTION PROCESS

**Who was involved in the data collection process (e.g., students, crowd workers, contractors)?** The annotations are provided by experienced annotators and multiple validation stages.

### B.6 USE

**What (other) tasks could the dataset be used for?** This dataset is primarily used for map association, a new task for online navigation refinement.

## C RELATED WORK

### C.1 END-TO-END AUTONOMOUS DRIVING

Planning, being an essential component for numerous fields Zhang et al. (2025a; 2024c; 2025b); Bai et al. (2024), is recognized as one of the pivotal research domains in the realm of autonomous driving at present. VAD Jiang et al. (2023) utilizes an ego query mechanism to forecast individual mode trajectories, whereas VADv2 Chen et al. (2024) enhances this by adopting a probabilistic framework that considers multiple trajectories. SparseDrive Sun et al. (2025) is innovative in its creation of parallel motion and planning modules, effectively decreasing the computational requirements of the BEV features. DiffusionDrive Liao et al. (2025) employs a truncated diffusion policy to improve the

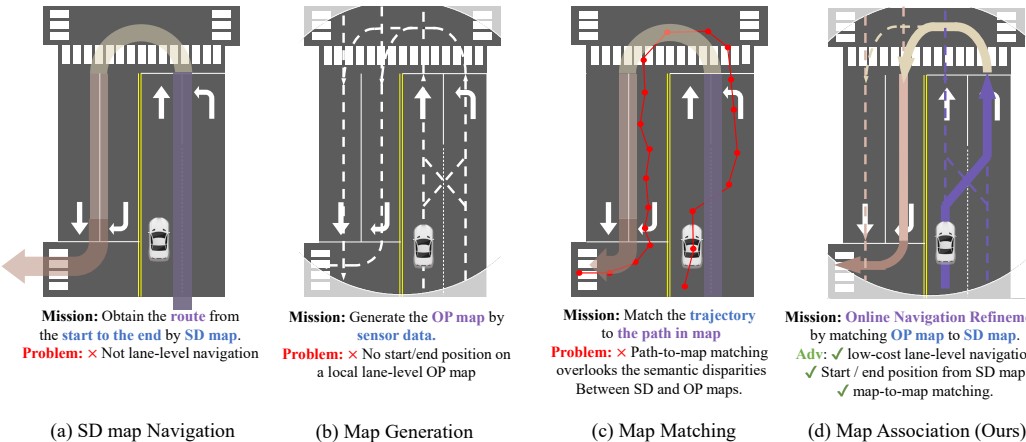

| (a) SD map Navigation | (b) Map Generation | (c) Map Matching | (d) Map Association (Ours) |

Figure 7: The comparison of (a) SD map navigation, (b) mapping generation, (c) map matching and (d) map association. The blue word is the input of the task and the purple word is the output of the task

probabilistic depiction of trajectories, while MomAD Song et al. (2025) aims to enhance stability and maintain consistency throughout sequential planning decisions. Significantly, unlike other end-to-end planning techniques, NavigScene Peng et al. (2025) uses navigation data based on SD maps, achieving more accurate and reliable autonomous driving, thus underscoring the significance of navigation data. Consequently, we propose that the refinement of lane-level navigation through MAT could further enhance the capabilities of current autonomous driving methodologies.

### C.2 MISSION COMPARISON

Fig. 7 compares four tasks: SD map navigation, map generation, map matching, and map association. Initially, navigation based on the standard definition (SD) map achieved a road-level path, which is now widely integrated into navigation apps. However, road-level navigation often does not provide precise directions for specific lanes and directions. As a result, the development of economical and quick lane-level navigation systems has emerged as a significant area of GIS. Next, lane-level online perception (OP) maps are generated using data from vehicle sensors by map generation. Unfortunately, such maps do not have a global topological framework, making it challenging to form a route from the start of the navigation to the endpoint. Subsequently, the task of map matching attempts to align GPS path with global SD or HD maps by following a path-to-map matching model. However, this model falls short when trying to match SD and OP maps due to disregarding the differences between these types. Finally, the map association task seeks to create linkages between SD and OP maps via a map-to-map association model. Using the SD map, the map association identifies a global route from the origin to the destination. Then this global route is meticulously translated into the OP map at the lane level through the association of the SD and OP maps, resulting in a comprehensive lane-level navigation path.

## D DATASET

### D.1 REVISED ANALYSIS.

As detailed in Section 4.1, the dataset is partitioned into OMA train, val, and test set, with statistics summarized in Tab. 7. The OMA training and val set comprises 26,111 training scenarios and 5,613 validation scenarios, totaling 31,724 samples, while the test set contains only 5,573 test scenarios due to the exclusion of low-quality predictions. Both data sets share identical spatial coverage, with HD maps covering $(\pm 15m, \pm 30m)$ and SD maps extending to $(\pm 75m, \pm 75m)$. Notably, the SD map's road density in OMA train and val set decreases from 15.1 roads/scene during training to 10.8 in validation/test splits, suggesting potential domain shifts between training and evaluation environments.

Table 7: Statistics of OMA train, val and test set.

| Split | Train | Val | Test |
|---|---|---|---|
| HD map Range | | $(\pm 15m, \pm 30m)$ | |
| SD map Range | | $(\pm 75m, \pm 75m)$ | |
| Scene Segment | 26111 | 5613 | 5573 |
| Avg. lane per scene | 81.3 | 75.8 | 310.34 |
| Avg. lane path per scene | 7.40 | 7.97 | 322.06 |
| Avg. boundary per scene | 3.48 | 3.31 | 9.65 |
| Avg. length per lane | 3.14m | 3.19m | 2.32m |
| Avg. length per boundary | 44.81m | 43.64m | 32.17m |
| Avg. road per scene | 15.1 | 10.8 | 10.8 |
| Avg. length per road | 38.2m | 50.3m | 50.3m |
| Avg. Connection per lane | 2.0 | 2.1 | 2.9 |
| Avg. Connection per road | 2.0 | 1.8 | 1.8 |
| Avg. Connection per boundary | 2.0 | 2.0 | 2.0 |

Table 8: Refinement statistics for different scenes in OMA

| Scene name | Number of centerlines | Number of refinements | Refinement ratio |
|---|---|---|---|
| Boston | 1,205,661 | 2,341 | 0.194% |
| Singapore | 910,140 | 3,682 | 0.404% |

Quantitative discrepancies between train, val and test set reveal systemic geometric and topological inconsistencies in predicted maps. The test set predicts an average of 310 lanes/scene, more than four times that of the Train and Val set (73.8), with significantly shorter mean lane lengths (2.32 m vs 3.16 m in trainval), indicating both over-segmentation and false positives. This fragmentation is further amplified by test's prediction of 322.06 lane paths/scene (vs. 7.69 in Train), where ground-truth lanes are frequently split into disconnected fragments. Boundaries exhibit similar degradation: test set detects 9.65 boundaries/scene (vs 3.40 in trainval) with reduced mean lengths (32.17m vs 44.23m), reflecting fragmented boundary detection. Meanwhile, validation data show slight degradation compared to training splits (e.g. 75.8 vs. 81.3 lanes/scene), highlighting inherent variability in real-world map quality.

Connectivity metrics expose deeper structural errors in test set predictions. The average lane connectivity in test set reaches 2.9, substantially higher than trainval's 2.0/2.1, revealing widespread mislinking of spatially disjoint lanes. Similarly, the validation data show reduced road connectivity (1.8 vs. 2.0 in training), suggesting a domain bias toward simpler topologies in training scenarios. Semantic associations between roads and lanes also degrade significantly. Training roads are associated with 1,547 lanes on average, collapsing to 945 in validation splits, which implies degradation of the hierarchical structure in complex scenarios.

These discrepancies have critical implications for benchmarking perception systems. The severe over-prediction and fragmentation in test set highlight the need for metrics penalizing false positives and disconnected paths (e.g., path-length-weighted scores). Furthermore, the mismatch between training and validation/test distributions (e.g., road count/length differences) necessitates domain adaptation strategies to ensure generalization. Finally, the collapse of semantic hierarchies in validation data suggests that end-to-end models may struggle to learn robust associations between roads and their constituent elements without explicit structural constraints. Together, these findings underscore the importance of a connectivity-aware association method to avoid overestimating performance on fragmented or mislinked predictions.

## D.2 RANGE OF SAMPLE

The model is trained on nuScenes using synchronized LiDAR and camera inputs with official configuration. For each sample, an SD map cropping measure $150\,m \times 150\,m$ centered around the ego vehicle preserves the adjacent topological context followed by the sensor setting of nuScenes Caesar et al. (2020a). For the OP map, the cropping measure of the OP map $30\,m \times 60\,m$ was centered around the vehicle of the ego, as referenced in Liao et al. (2023b); Liu et al. (2023a); Li et al. (2022).

## D.3 PON SPLIT AND OP MAPS

Both Train, val and test set apply the pon split Roddick & Cipolla (2020) of the nuScenes dataset Caesar et al. (2020b), ensuring that there is no leakage between the training and validation datasets. For consistent lane prediction segmentation with OMA test set, we re-trained MapTRv2 Liao et al. (2023b) using the nuScenes dataset with the pon split. Drawing inspiration from the private protocol in MOT17/MOT20 Dendorfer et al. (2021), we suggest that future research evaluates the test set with an enhanced centerline prediction network, without relying on MapTRv2 as a baseline. Furthermore, we will establish an open evaluation protocol allowing submissions to include their own OP maps, thereby reducing reliance on a single model.

## D.4 GEOGRAPHICAL GENERALIZATION

Regarding the geographical generalization of OMA, we elaborate from two aspects: First, OMA's data originates from the nuScenes dataset, which is widely used in the autonomous driving field and contains data from two different countries, Singapore and Boston. Given that nuScenes has become a core dataset in the autonomous driving field for perception, mapping, and planning since its introduction, we believe that the nuScenes data itself possesses certain representativeness and generalizability. Second, we conducted cross-validation ablation experiments in the main text. The cross-experimental results of MAT on OMA demonstrate that MAT models trained on different geographical regions possess certain cross-regional generalization capabilities, reflecting the good geographical generalizability of the OMA dataset itself.

## D.5 ANNOTATION WORKFLOW

The OMA data annotation process is divided into two primary stages: First, we performed coarse alignment based on the GPS coordinates from the SD map of OSM and the GT OP map of NuScenes. Second, we employed a skilled annotation team to correlate the SD map with the real perception map. The specialists were then engaged to evaluate and improve the annotations, ensuring their ultimate quality. Tab. 8 presents the total count and percentage of data changes in various regions in Phase 3. It is evident that the alteration rate for Singapore and Boston remained less than 1%, suggesting a generally high standard of data annotation quality.

## D.6 HANDLING TRANSITION AMBIGUITY

Addressing the concern regarding transition lines, we acknowledge that assigning roads in such transition areas presents inherent ambiguity due to the heterogeneity between SD and OP Maps. To mitigate potential impacts on training stability and evaluation fairness, we implemented specific strategies: (1) *Standardized Annotation Protocol:* We established a unified rule where the assignment of a Lane Vector at an SD Link transition boundary is determined by the SD Link closest to the Lane Vector's start point, ensuring topological consistency. (2) *Tolerance in Evaluation Metrics:* Our metric assesses the precision of path-level associations across 11 thresholds (50% to 95%). Crucially, the 5% tolerance buffer included even at the strictest threshold is specifically designed to accommodate inevitable semantic ambiguity at transition boundaries.

## D.7 FUTURE EXPANSION STRATEGY

We acknowledge the importance of scaling the benchmark to encompass more diverse datasets (e.g., Argoverse). To minimize annotation costs during future expansion, we propose a "Human-in-the-Loop" iterative annotation strategy. This approach is strongly supported by our data efficiency

experiments as shown in Tab. 6, which demonstrated that MAT can achieve competitive performance with as little as 5% of the training data. The specific pipeline is as follows:

1. **Model-Assisted Pre-annotation:** We utilize the MAT model trained on the existing OMA (nuScenes) dataset to perform zero-shot or few-shot inference on unlabeled new data. This generates initial "draft" associations (e.g., SD-to-HD correspondences and topology) and significantly reduces the cold-start problem.

2. **Lightweight Manual Correction:** Instead of labeling from scratch, annotators focus solely on verifying and correcting the model's high-confidence predictions. Given the model's high data efficiency, the pre-annotation quality improves rapidly even with a small set of initial corrections.

3. **Closed-Loop Iteration:** The corrected data is immediately integrated into the training set to fine-tune the model. The updated model is then used to pre-annotate the subsequent batch of data, creating a positive feedback loop that progressively reduces manual workload.

# E    METRIC

## E.1    DETAILS OF NR-PR

In the main article, we present a narrative explanation of the Navigation Refinement P-R accompanied by a schematic diagram. To elucidate the calculation of Navigation Refinement P-R more thoroughly, we include the pseudo-code for computing Navigation Refinement P-R, as depicted in Alg. 1.

Furthermore, the formula for NR-P$^{50:95}$, NR-R$^{50:95}$ and NR-F1$^{th}$, NR-P$^{50:95}$is as follows:

$$\text{NR-P}^{50:95} = \sum_{th \in T} \text{NR-P}^{th}, \quad \text{NR-R}^{50:95} = \sum_{th \in T} \text{NR-R}^{th}$$
$$\text{NR-F1}^{th} = \frac{2\text{NR-P}^{th} \cdot \text{NR-R}^{th}}{\text{NR-P}^{th} + \text{NR-R}^{th}}, \quad \text{NR-F1}^{50:95} = \frac{2\text{NR-P}^{50:95} \cdot \text{NR-R}^{50:95}}{\text{NR-P}^{50:95} + \text{NR-R}^{50:95}} \quad (7)$$

where $th$ are the thresholds in association P-R as $[0.5 : 0.05 : 0.95]$ (10 thresholds). Additionally, in the validation set, because NR-R is always 1, we use the NR-P value directly as NR-F1.

## E.2    CURVATURE-BASED WEIGHTED METRIC

Although the length-interval stratification in the proposed NR-PR metric effectively mitigates bias towards short paths, it does not explicitly account for the geometric complexity of roads. In real-world datasets, straight roads are statistically more frequent than complex curved roads, potentially masking model deficiencies in handling high-curvature topologies. To address this and evaluate the model's fairness across diverse road types, we introduce a **Curvature-based Weighted Metric**.

**Complexity Measure.** Defining complexity via average angular changes can be highly sensitive to point sampling rates. Therefore, we employ the **Discrete Fréchet Distance** between the actual path and the straight line connecting its endpoints as a robust measure of geometric complexity. A higher Fréchet distance indicates a greater deviation from a straight line, representing higher curvature or irregularity.

**Stratification Strategy.** As shown in Fig. 8, similar to length-based approach, we adopt a stratified statistical method to handle the long-tailed distribution of road complexity. We identify the 95th percentile of complexity scores in the dataset as an upper bound and uniformly divide the range $[0, 95\text{th percentile}]$ into 10 bins. The final metric is computed by averaging the NR-F1 scores across these bins, ensuring that complex road geometries contribute equally to the final score, rather than being overshadowed by the dominant straight roads.

**Quantitative Analysis.** We evaluated the MAT model on the OMA dataset using this weighted metric. Table 9 illustrates the impact of stratification on the overall score. As the number of bins increases—forcing the metric to weigh complex roads equally to straight ones—the overall

---

**Algorithm 1** Evaluate Navigation Refinement P-R

---

1: **function** EVALMETRIC(pred_centerline, gt_centerline, threshold, acc_list)
2:     **Input:** pred_centerline, gt_centerline, threshold, acc_list
3:     **Step 1: Point Matching**
4:     EXTRACTPOINTS(pred_centerline)
5:     EXTRACTPOINTS(gt_centerline)
6:     POINTMATCH(pred_sample, gt_sample_point, threshold)
7:     **Step 2: Path Matching**
8:     INITIALIZECOUNTERS(TP, FP, FN, acc_list)
9:     **for all** point pairs $(i, j)$ in matched points **do**
10:         Find pred_path and gt_path between points $i, j$
11:         PATHMATCH(pred_path, gt_path, threshold)
12:         **if** paths match **then**
13:             Check sequence consistency and accuracy
14:             **for all** $acc \in acc\_list$ **do**
15:                 Update TP/FP based on accuracy vs $acc$
16:             **end for**
17:         **end if**
18:     **end for**
19:     **Step 3: Count Unmatched Paths**
20:     **for all** unmatched gt path **do**
21:         **for all** $acc \in acc\_list$ **do**
22:             $FN[acc][k] \leftarrow FN[acc][k] + 1$
23:         **end for**
24:     **end for**
25:     **Step 4: Calculate Precision and Recall**
26:     **Initialize:** $Precision \leftarrow \{\}, Recall \leftarrow \{\}$
27:     **for all** $acc \in acc\_list$ **do**
28:         $denominator\_p \leftarrow TP[acc] + FP[acc]$
29:         $denominator\_r \leftarrow TP[acc] + FN[acc]$
30:         $Precision[acc] \leftarrow \begin{cases} TP[acc]/denominator\_p & \text{if } denominator\_p > 0 \\ 0 & \text{otherwise} \end{cases}$
31:         $Recall[acc] \leftarrow \begin{cases} TP[acc]/denominator\_r & \text{if } denominator\_r > 0 \\ 0 & \text{otherwise} \end{cases}$
32:     **end for**
33:     **Return:** TP, FP, FN, Precision, Recall
34: **end function**

---

Table 9: Impact of Complexity Stratification on NR-F1. The metric score decreases as we enforce equal weighting for complex road types (increasing $N$), revealing the dominance of simple roads in the unweighted score.

| Complexity Bins ($N$) | 1 | 2 | 5 | 10 |
|---|---|---|---|---|
| Val Set | 81.6 | 70.2 | 62.7 | 60.3 |
| Test Set | 51.8 | 38.7 | 33.5 | 32.5 |

performance drops significantly (e.g., from 81.6% to 60.3% on the Val set). This confirms that the dataset is dominated by simple geometries where the model performs exceptionally well, masking the challenges posed by complex topologies in unweighted metrics.

Table 10 details the performance in specific complexity intervals. The results reveal a strong negative correlation between road complexity and association precision. On simple straight roads (Bin 1), the model achieves high precision (87.2% on Val). However, performance degrades drastically on high-complexity paths (dropping to 43.1% in Bin 10). This trend is even more pronounced in the Test set, indicating that while MAT is robust, geometric complexity remains a significant challenge

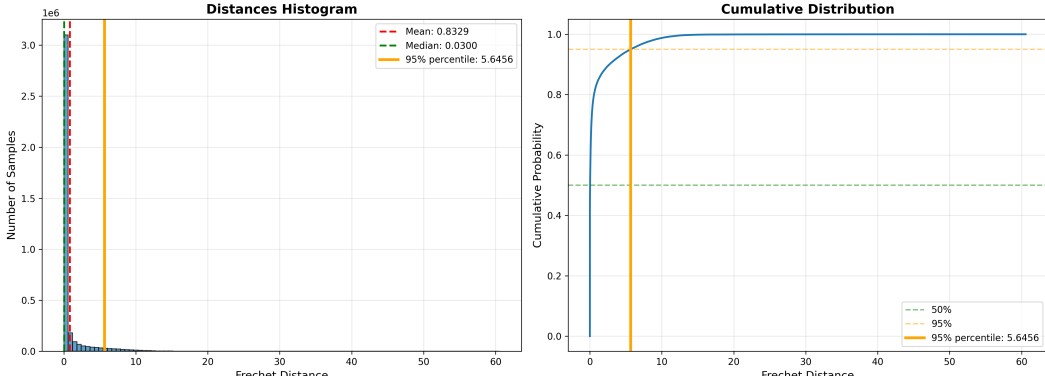

Figure 8: Statistics of Fréchet distances across all paths in the OMA dataset. **(1)** Histogram of Fréchet distances. A substantial number of paths exhibit Fréchet distances concentrated within the range of $[0, 1]$, revealing a pronounced long-tail distribution in the overall data. **(2)** Cumulative probability of Fréchet distances. Consistent with (1), the distribution displays a significant long-tail characteristic, underscoring the importance of block-wise statistics for the NR-PR metric.

Table 10: Performance per Complexity Bin ($N = 10$). Bin 1 represents straight roads, while Bin 10 represents high-curvature/irregular roads. The results indicate a performance degradation as geometric complexity increases.

| Bin Index | 1 | 2 | 3 | 4 | 5 | 6 | 7 | 8 | 9 | 10 |
|-----------|------|------|------|------|------|------|------|------|------|------|
| Val Set   | 87.2 | 63.2 | 56.7 | 50.7 | 49.9 | 49.1 | 46.5 | 45.2 | 42.7 | 43.1 |
| Test Set  | 57.9 | 50.4 | 32.0 | 21.2 | 16.8 | 14.7 | 10.7 | 10.6 | 10.4 | 11.6 |

for map association tasks. This analysis serves as a complement to the length-based metric, offering a more comprehensive view of the robustness of the model.

## F    METHOD

In this section, we delve deeper into the technical aspects of the model, including Path-aware attention, spatial attention, and the model's post-processing.

### F.1    PATH-AWARE ATTENTION

The particular design of path-aware attention (PA) can be seen in Fig 9 (a). The fundamental framework of PA is made up of four components: computing order and its inverse, reorganizing tokens, calculating attention, and inverting tokens.

Path-aware attention uses paths to determine the sequence of tokens. Initially, we define the network of roads or centerlines and then identify all complete paths from a starting point (with no incoming connections) to an endpoint (with no outgoing connections). We concatenate these paths to generate a path-based token sequence. During the reordering of path-aware attention, a token may appear across various paths simultaneously, requiring us to duplicate the token. Then, we compute attention by segregating tokens based on their paths, ensuring that interactions occur only among tokens within the same path. After attention calculation, the tokens are reversed to match the original input sequence. If multiple tokens exist within the path of a single original token, they are averaged.

It should be highlighted that the model organizes paths internally to prevent overly lengthy paths. If a path surpasses the patch size, it is divided into several groups, where each group (aside from the final one) matches the patch size. PA's worst-case time complexity is $O(kn^3)$, where $k$ is the patch size and $n$ is the number of tokens, if and only if $\frac{2}{3}$ centerline or road are either starting points or ending points. Clearly, this is highly improbable in real-world scenarios, and its average time complexity remains $O(k^2n)$, which is linear.

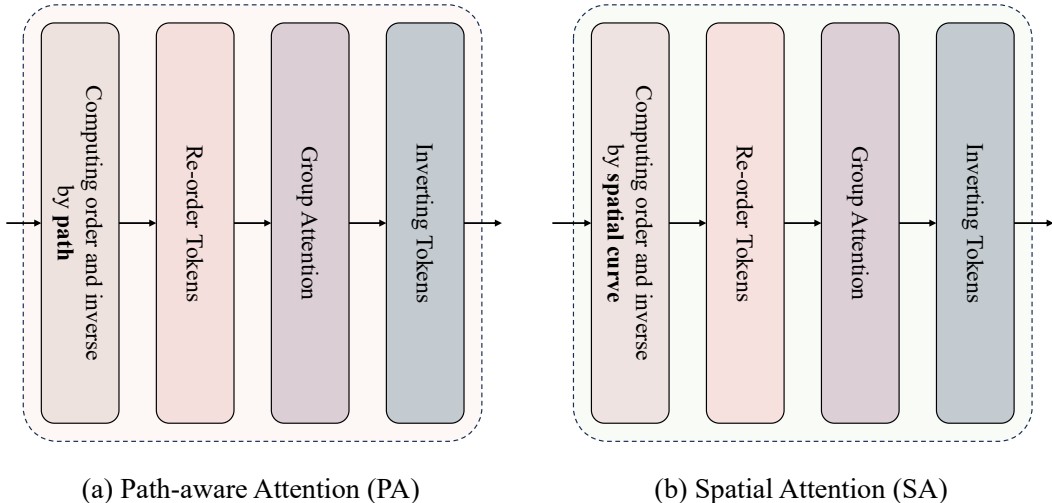

(a) Path-aware Attention (PA)    (b) Spatial Attention (SA)

Figure 9: Overview of Path-aware attention and spatial attention.

## F.2 SPATIAL ATTENTION

Fig. 9 (b) provides a detailed description of the architecture of the SA model. Similarly to pa, the fundamental structure of SA includes four main components: computing sorting and its reverse, rearranging tokens according to the sorted order, calculating attention, and then reverse sorting the tokens again.

In order of SA, we apply a space filling curve $\varphi^{-1} : \mathbb{Z}^3 \to \mathbb{Z}$ to serialize the vectors in a 1D sequence, preserving spatial locality. Four curves are used: Z-order, Transposed-Z, Hilbert, and Transposed-Hilbert. To avoid bias toward specific curve types, we randomly select one curve per training iteration. In addition, similar to PA, if there are multiple tokens that belong to a single token after the coordinate calculation, we will average the multiple tokens in the sort and copy that token to all the corresponding tokens in the reverse sort.

During the group stage, tokens are partitioned based on the sorted sequence, each partition matching the patch size. Following this, the model executes self-attention computations within these partitions. Similar to PA, the self-attention (SA) algorithm has a time complexity of $O(kn)$, where $k$ is the patch size and $n$ is the number of token, indicating that the complexity of SA remains linear.

## F.3 POST PROCESS

In the main manuscript, we offer a narrative explanation of the post-processing. To enhance clarity, we also present a mathematical formulation of the post-processing details. Let $\mathcal{R}$ denote the road as the vocabulary in the traditional beam search with size $|\mathcal{R}|$, and let $k = 4$ represent the width of the beam. At each step $t$, the algorithm maintains a set $B_t$ of candidates path $k$, each associated with a score $s(h)$ defined as the sum of logarithmic conditional probabilities. The search begins by selecting the token $w^*$ with the maximum initial probability $P(w|x)$ given as input $x$, forming the singleton initial set:

$$B_0 = \text{Top}_1 \left( \mathcal{R}, \ \log P(w|x) \right), \tag{8}$$

which simplifies to:

$$B_0 = \{[w^*]\}, \ \text{where} \ \log P(w^*|x) = \max_{w \in \mathcal{R}} \log P(w|x). \tag{9}$$

This initialization bypasses conventional fixed start tokens and prioritizes high-probability seeds.

In iteration $t \geq 1$, each sequence $h \in B_{t-1}$ generates $2|\mathcal{R}|$ candidates by appending a token $w \in \mathcal{R}$ to the left $(w \cdot h)$ or right $(h \cdot w)$ of $h$, forming the expanded candidate set:

$$\mathcal{C}_t = \{w \cdot h \mid h \in B_{t-1}, \ w \in \mathcal{R}\} \cup \{h \cdot w \mid h \in B_{t-1}, \ w \in \mathcal{R}\}. \tag{10}$$

Table 11: Ablation study of patch size of PA and SA.

| Patch Size | 64 | 256 | 1024 | 2048 | $\infty$ |
|---|---|---|---|---|---|
| NR-F1$^{50:95}$ | 77.8 | 78.4 | **78.7** | 78.5 | 78.4 |
| Latency/ms | 77 | **68** | 70 | 72 | 72 |

Table 12: Ablation study of group method of PA.

| Group Method | N.G | Category | Path |
|---|---|---|---|
| NR-F1$^{50:95}$ | 78.0 | 78.2 | **78.7** |
| Latency/ms | 75 | 73 | **70** |

Table 13: Ablation study of Input size of SD Map

| Input size of SD Map | NR-F1$^{50:95}$ | Latency/ms |
|---|---|---|
| $60 \times 60$m | 78.6 | 67 |
| $90 \times 90$m | 78.6 | 69 |
| $120 \times 120$m | 78.7 | 69 |
| $150 \times 150$m | 78.7 | 70 |

h extension updates the sequence score using direction-specific conditional probabilities:

$$s(h') = \begin{cases} s(h) + \log P(w|h, \text{left}, x), & \text{if } h' = w \cdot h \\ s(h) + \log P(w|h, \text{right}, x), & \text{if } h' = h \cdot w \end{cases}. \tag{11}$$

The top-$k$ candidates from $\mathcal{C}_t$ are retained to form $B_t$:

$$B_t = \text{Top}_k \left( \mathcal{C}_t, \ s(\cdot) \right), \tag{12}$$

i.e.,

$$B_t = \{h'_1, h'_2, \ldots, h'_k\}, \text{ with } s(h'_1) \geq s(h'_2) \geq \cdots \geq s(h'_k). \tag{13}$$

The process ends at a predefined maximum length $T$ or when all sequences emit an end-of-sequence token, with the final output $\hat{h}$ selected as:

$$\hat{h} = \arg \max_{h \in \bigcup_{t=0}^{T} B_t} s(h). \tag{14}$$

## G   ABLATION STUDY

For the hyperparameters we proposed, additional ablation studies were executed to exhibit their stability.

### G.1   HYPER PARAMETER

**Patch Size of PA and SA.** Tab. 11 shows that precision plateaus at patch size 256 (NR-F1$^{50:95}$ = 78.4%) but increases slightly at 1024 (+0.3%) with a latency trade-off (+2 ms).

**Group Method of PA.** For PA grouping (Tab. 12), path-based grouping surpasses non-grouping (+0.7%) and category-based baselines (+0.5%), likely due to reduced cross-path interference.

**Input size of SD map.** Table 13 presents the results of the adjustment of the SD map input size. The findings reveal that alterations in the SD input size exert minimal influence on NR-F1 ($\leq 0.1$).

**Sampling size of SA.** Table 14 shows the findings for the spatial sample size $\delta$. When $\delta$ is set at 1m, there is a drop in model accuracy, which may be attributed to the challenge of differentiating nearby lane lines with such a substantial sampling size. When $\delta$ is below 0.5m, the accuracy of the model is decreased ($< 0.6$). Decreasing the interval further to 0.01 does not improve the accuracy of the model.

**Hyperparameter of loss function.** As illustrated in Table 15, the ablation studies for $\beta_{ce}$ and $\beta_{ctc}$ indicate that high beta values make CTC the main loss, which reduces the precision of the model. In contrast, lower beta values allow CE loss to predominate, leading to stabilized NR-F1 ($< 0.3$).

Table 14: Ablation study of $\delta$

| $\delta$ | NR-F1$^{50:95}$ | Latency |
|---|---|---|
| 1m | 74.6 | 65 |
| 0.5m | 78.1 | 68 |
| 0.1m | 78.7 | 70 |
| 0.01m | 78.7 | 79 |

Table 15: Ablation study of $\beta_{ce}$ and $\beta_{ctc}$ in loss function

| $\beta_{ce} : \beta_{ctc}$ | NR-F1$^{50:95}$ | Latency |
|---|---|---|
| 1:1 | 68.4 | 70 |
| 1:0.1 | 78.5 | 70 |
| 1:0.01 | 78.7 | 70 |
| 1:0.001 | 78.4 | 70 |
| 1:0 | 78.4 | 70 |

Table 16: Ablation study of space curve in SA.

| Spatial Curve | NR-F1$^{50:95}$ | Latency/ms |
|---|---|---|
| Z | 77.6 | 70 |
| Z + TZ | 78.1 | 70 |
| H + TH | 78.2 | 70 |
| Z + TZ + H + TH | 78.7 | 70 |

Table 17: Ablation study of beam width $k$ in post process.

| Beam Width | NR-F1$^{50:95}$ | Latency/ms |
|---|---|---|
| 1 | 78.2 | 69 |
| 2 | 78.6 | 70 |
| 4 | 78.7 | 70 |
| 8 | 78.7 | 78 |

**Space Curve of SA.** Table 16 displays the result of ablation studies focusing on spatial curves in SA. The findings suggest that adding more diverse spatial curves improves the model's performance. It is crucial to emphasize that including multiple spatial curve types does not affect model latency, given that each layer utilizes only a single spatial curve. Thus, the variety of spatial curve types does not increase the frequency of spatial curve sorting.

**Beam width of post-process.** Table 17 shows the results of the ablation test regarding the hyperparameter of beam width $k$ during post-processing. The study reveals that setting $k$ to 4 results in the best trade-off between accuracy and processing time.

**Length intervals of NR P-R.** To further investigate the influence of the path length distribution on the NR-F1 metric, we performed a stratified evaluation based on length intervals. The motivation for introducing length intervals stems from our observation that within the dataset, short-distance paths occur far more frequently than long-distance ones, and their corresponding F1 scores tend to be higher. Without appropriate handling, this imbalance could bias the overall NR-F1 score toward performance on short-distance paths. Inspired by the size-based categorization strategy adopted in the MS COCO benchmark Lin et al. (2014), we categorize paths into discrete length intervals, compute the F1 score within each interval, and subsequently aggregate the results into a unified NR-F1 measure. This procedure ensures that each length range contributes proportionally to the final metric, thus mitigating the dominance of short-distance paths in the evaluation. As shown in Tab. 18, omitting this stratification (that is, computing the metric on all paths without length-based grouping) produces a substantially inflated overall score. In contrast, increasing the number of length intervals causes the aggregated metric to gradually decrease and eventually converge to a stable value. These findings suggest that the length interval-based computation allows NR-F1 to provide a more balanced and fair evaluation on both the short- and long-distance paths, resulting in a more comprehensive evaluation of the model's ability to capture relational associations.

**Distance Threshold of NR P-R.** Tab. 19 highlights the effect of various distance thresholds on the final metrics in NR P-R Step 1. These thresholds originate from those used in Reachability P-R. It is important to note that Val uses the ground-truth OP map for predictions, thus all roads are deemed true roads, while the distance threshold applies solely to the test set. Ablation experiments reveal that distance thresholds significantly influence the accuracy outcome of NR-F1. With a strict threshold of 0.5m, most of the lane lines on the OP map are regarded false detections, as they do not meet the requirement. In contrast, a more lenient threshold of 2.0 m allows most lane lines to be perceived as correct, bringing the NR-F1 to a value comparable to that of the Val set. This also indicates that there is no domain difference between the Test and Val sets; discrepancies in results are due to the threshold settings. However, considering that the accuracy of the lane regression is also pivotal, we opted for a balanced threshold of 1.0m to judge whether the OP map correctly represents a lane.

Table 18: Ablation study of length intervals of NR P-R.

| Number of length intervals | 1 | 2 | 3 | 6 | 10 | 15 |
|---|---|---|---|---|---|---|
| NR-F1$^{50:95}$ / Val | 81.8 | 79.0 | 78.5 | 78.2 | 78.7 | 78.7 |
| NR-F1$^{50:95}$ / Test | 51.6 | 47.6 | 47.2 | 45.3 | 44.9 | 45.0 |

Table 19: Ablation study of distance threshold of NR P-R.

| Distance threshold | 0.5m | 1.0m | 1.5m | 2.0m |
|---|---|---|---|---|
| NR-F1$^{50:95}$ / Val | 78.7 | 78.7 | 78.7 | 78.7 |
| NR-F1$^{50:95}$ / Test | 6.6 | 45.0 | 68.2 | 72.4 |

## G.2 ROBUSTNESS TO IMPERFECT SD MAPS

In real-world applications, Standard Definition (SD) maps often suffer from imperfections caused by GPS localization errors, sensor noise, or map construction delays. To quantitatively evaluate the robustness of MAT and the proposed metrics against such imperfect inputs, we conducted additional ablation studies on the OMA Validation set.

We categorized SD map imperfections into three distinct types and injected varying levels of noise (Low, Medium, High) into the input SD maps. The noise generation protocols are defined as follows:

- **Global Shift:** Simulates GPS localization errors by applying a uniform distribution offset relative to the global map range. We tested noise ratios of 10%, 20%, and 50%. Crucially, for each SD map sample, the same random offset vector is applied to all road vectors, preserving the internal shape but shifting the absolute position.

- **Element Noise:** Simulates geometric construction errors by applying random perturbations relative to the global map range. We tested noise ratios of 5%, 10%, and 20%. Unlike Global Shift, the noise here is generated independently for each vector within the SD map, resulting in geometric jitter and shape distortion.

- **Element Absence:** Simulates map construction incompleteness by randomly masking out road vectors. We evaluated omission ratios of 10%, 20%, and 30%, where vectors are randomly discarded from the SD map input.

The results, measured by the NR-F1$^{50:95}$ score, are summarized in Tab. 20. The results demonstrate the resilience of the proposed method under varying noise conditions:

- **Robustness to Shift:** Even under a high global shift of 50% , the model maintains reasonable performance (74.6 / 41.4). This indicates that our **Path-Aware Attention (PA)** successfully captures the relative topological structure of the road network, reducing dependency on absolute coordinate alignment.

- **Robustness to Element Noise:** The performance remains highly stable, dropping only marginally from 78.3 / 44.8 to 77.5 / 44.1 despite a 20% geometric jitter. This suggests that the **Spatial Attention (SA)** mechanism effectively aggregates global contextual information, thereby mitigating the impact of local geometric inaccuracies.

- **Robustness to Missing Elements:** The model exhibits strong adaptability to missing data, maintaining a score of 75.5 / 43.1 even when 30% of the SD map vectors are absent. The Transformer architecture effectively infers associations based on the remaining context and the overall graph topology, ensuring that the association process does not fail catastrophically when individual segments are missing.

These experiments confirm that MAT maintains strong robustness when dealing with imperfect SD maps, effectively handling geometric misalignment, shape distortion, and topological incompleteness inherent in real-world navigation tasks.

Table 20: Ablation study of robustness to imperfect SD Maps on OMA Val / Test set with varying noise intensities. The metric reported is NR-F1$^{50:95}$.

| Noise Type | Low | Medium | High |
|---|---|---|---|
| Global Shift (10%, 20%, 50%) | 77.5 / 43.3 | 76.0 / 42.9 | 74.6 / 41.4 |
| Element Noise (5%, 10%, 20%) | 78.3 / 44.8 | 77.5 / 44.3 | 77.5 / 44.1 |
| Element Absence (10%, 12%, 30%) | 77.5 / 44.0 | 76.4 / 43.7 | 75.5 / 43.1 |

Table 21: Zero-shot robustness evaluation on nuScenes-C under different environmental corruptions.

| Condition | Easy | Mid | Hard |
|---|---|---|---|
| Rain | 44.7 | 42.1 | 40.6 |
| Fog | 44.2 | 43.4 | 42.1 |
| Dark | 44.0 | 43.8 | 41.6 |

### G.3    ROBUSTNESS UNDER ENVIRONMENTAL NOISE AND OCCLUSION

To evaluate the robustness of MAT against sensor failures caused by environmental factors and severe occlusions, we conducted extended experiments focusing on two aspects: quantitative evaluation on corrupted inputs and qualitative analysis of occlusion scenarios.

**Robustness against Environmental Noise.** We utilized *nuScenes-C* Xie et al. (2025b), a benchmark designed to assess robustness against input corruptions, to simulate sensor degradation. Specifically, we used MapTRv2 Liao et al. (2023b) to perform inference on the *nuScenes-C* validation set with pon split, generating OP maps under three specific weather conditions: **Rain**, **Fog** and **Dark** (Night). These conditions were categorized into three difficulty levels: Easy, medium, and Hard followed by the nuScenes-C setting.

Crucially, we adopted a **zero-shot inference** setting: the MAT model used for association was trained solely on clean data and was **not retrained** on these corrupted samples. This setting serves as a rigorous stress test for the generalizability of the model.

Table 21 reports the quantitative results with NR-F1$^{50:95}$ score. As observed, MAT maintains robust performance across all conditions. Even in "hard" settings, where visual features are significantly compromised, the performance drop is minimal. For example, in Fog scenarios, the metric only decreases from 44.2 (Easy) to 42.1 (Hard). This stability demonstrates that, by taking advantage of the topological priors of SD maps, MAT effectively mitigates the impact of sensor noise.

**Visualization of Environmental Degradation.** To intuitively display the challenge, Fig. 10 illustrates the synthesized degradation samples of the OP maps at different difficulty levels. Fig. 11 and Fig. 12 further compare the inference results under the "Hard" setting. Despite the severe noise in the input OP maps (e.g. missing boundaries or phantom lanes due to fog/rain), MAT successfully recovers the correct lane topology by aligning it with the SD map.

**Analysis of Severe Occlusion.** While the explicit quantitative evaluation of occlusion is constrained by the lack of ground-truth annotations, we analyze it physically. Vehicle occlusion typically manifests itself in OP maps as lane break or missing segments. Since MAT is explicitly designed to model global topology, it is inherently robust to such defects.

Fig. 12 presents a case study of **severe vehicle occlusion**. As shown in the visualization, MapTRv2 manages to reconstruct a coherent global road structure despite significant visual blockage by vehicles. We attribute this capability to the temporal modeling design of the upstream perception model (MapTRv2), where single-frame occlusion does not disrupt the map modeling derived from the overall temporal sequence. Taking advantage of the inherent robustness of MapTRv2 against occlusion, our downstream MAT model is consequently largely unaffected by dynamic vehicle occlusion, ensuring stable association performance.

(a) None

(b) Easy

(c) Mid

(d) Hard

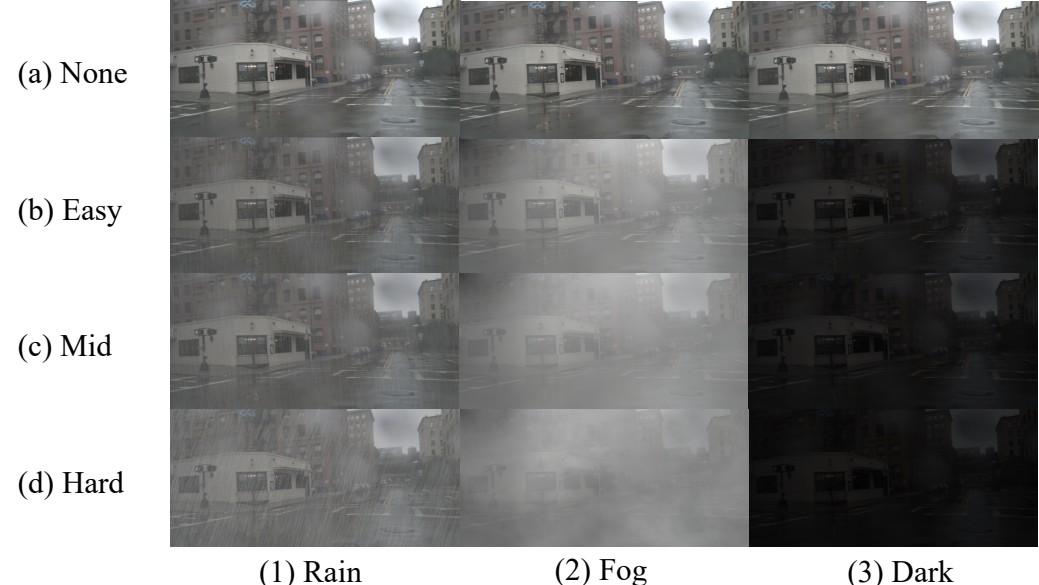

(1) Rain      (2) Fog      (3) Dark

Figure 10: **Visualization of OP map degradation samples from nuScenes-C.** The columns represent different environmental conditions (Rain, Fog, Dark), and the rows represent increasing difficulty levels (Easy, Mid, Hard). These noisy inputs simulate realistic sensor failures.

## H  VISUALIZATION

This section presents the visualization of our model, featuring path-aware attention (PA) and spatial attention (SA) alongside the model's results. Furthermore, we examine the failed case with an analysis of our model.

### H.1  ATTENTION MAP

The upper portion of Fig.13 presents visualizations of Spatial Attention (SA) maps at different stages of the model. As revealed by the analysis, SA provides extensive receptive fields in the early stages, enabling tokens to capture global contextual information. Specifically, during Stage 1 and Stage 2, the SA attention distributions exhibit highly dispersed patterns, allowing each query token to uniformly attend to global regions across the input space. In later stages, the functional role of SA transitions to facilitating cross-category token interactions. For example, in Stages 3-5, distinct attention patterns emerge where tokens primarily interact with their semantically corresponding road elements. Notably, this interaction is not strictly confined to the road tokens directly associated with the centerline token - significant attention weights also develop between the centerline token and adjacent road segments. As exemplified in Stage 4, the tokens establish prominent attention links with multiple road tokens along the same path. We posit that this expanded interaction mechanism constitutes a critical component for precise centerline localization. The propagation of attention observed in later stages effectively enables the maintenance of geometric coherence between spatially distributed road elements while preserving discriminative semantic information through long-range dependencies.

In contrast, the lower part of Fig.13 visualizes the Path-aware Attention (PA) maps at different stages of the model. The visualization reveals that PA primarily focuses on neighboring tokens adjacent to the target path tokens, effectively serving as a local information extractor. Experimental results demonstrate that this localized information extraction capability plays a pivotal role in the model performance, exhibiting a marked contrast with the global perception mechanism of SA. We posit that SA specializes in capturing global contextual patterns while PA emphasizes localized feature extraction. This dual-attention paradigm establishes a synergistic interplay between global and local perception, achieving an optimal balance between comprehensive understanding and fine-grained detail processing, thereby substantially enhancing the model's overall effectiveness.

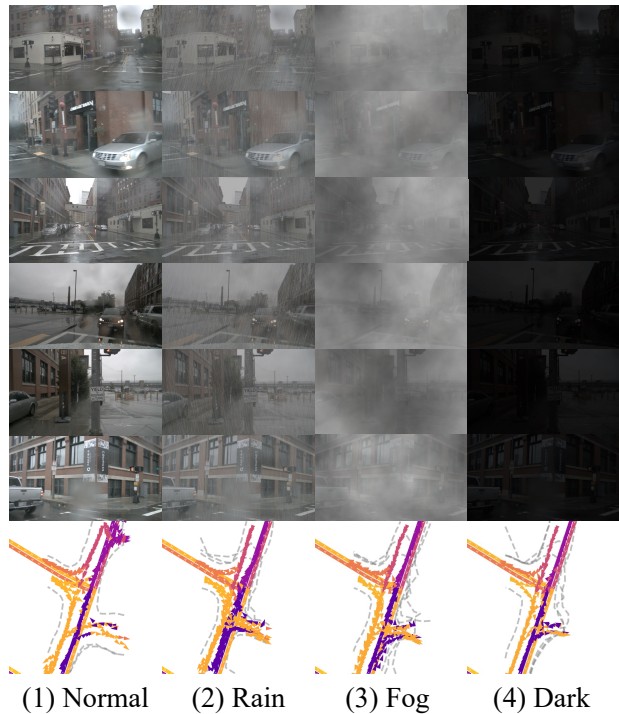

(1) Normal    (2) Rain    (3) Fog    (4) Dark

Figure 11: **Qualitative results under "Hard" environmental conditions.** The figure compares the ground truth (GT) with MAT's predictions under severe Rain, Fog, and Dark conditions. The model demonstrates strong zero-shot robustness, effectively associating noisy OP lanes with SD paths.

## H.2 RESULT COMPARE

Fig.14 illustrates comparisons of model ground truth on KNN, HMM, MAT and MAT w/ postprocess. The top rows of (d) and (e) exhibit our post-processing module's effectiveness. MAT predictions initially display incorrect topological connections where roads are mistakenly linked (marked with red circles). Our postprocessing, which utilizes topology-aware beam searching, rectifies this by eliminating non-sequential transitions and reconstructing precise topological paths. The second row demonstrates MAT's superior handling of complex topologies. Although HMM targets sequential path associations, its single path paradigm often underperforms in complex topologies with intersections. In contrast, our model uses spatial attention to grasp global information and cross-path associations, facilitating adaptive learning of complex topological patterns for accurate connectivity inference. The third row showcases our model's improved ability to localize associations. Using path-aware attention, the model emphasizes detailed extraction of local features along paths. This targeted local perception ensures precise associations at challenging points, such as junctions, where HMM is typically short due to limited contextual understanding.

Fig. 15 illustrates a comparison of ground truth results for KNN, HMM, MAT, and MAT with postprocessing. Significantly, the visualization demonstrates that our model excels in map association in noisy scenarios with inaccurate centerline predictions, surpassing KNN and HMM by integrating the complementary benefits of global association (SA) and local detail refinement (PA).

## H.3 FAILED CASES

Fig. 16 illustrates the failed cases of the MAT. Our study reveals that the key challenge is the localization errors associated with spatial misalignment between the predicted paths and the actual labels. This discrepancy significantly affects the accuracy of the association, particularly at critical junctures where complex path interactions create ambiguous topological patterns. Although all baseline methods exhibit substantial association errors under these difficult conditions, our model achieves notable error reduction due to its dual-attention framework. However, discrepancies between

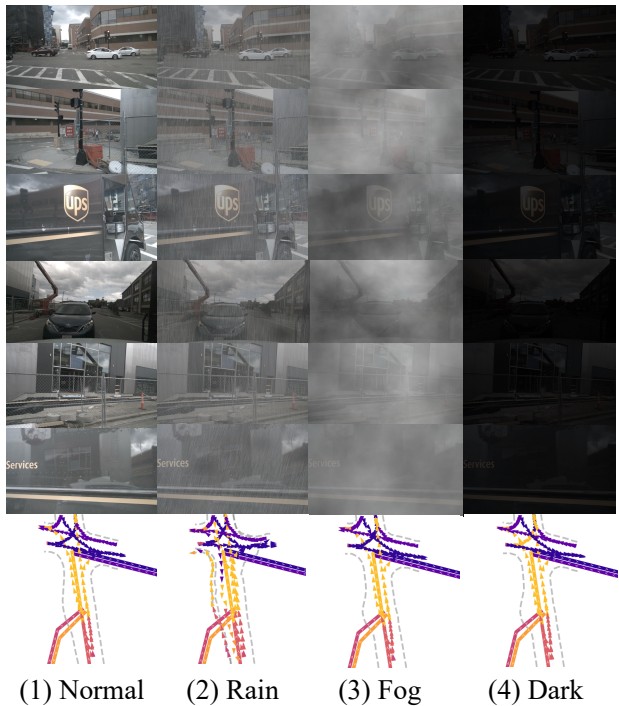

(1) Normal (2) Rain (3) Fog (4) Dark

Figure 12: **Qualitative results under "Hard" environmental conditions and severe occlusion.** Similar to the setup in Fig. 10, this figure displays the inference results under Rain, Fog, and Dark scenarios, but specifically highlights a case of severe vehicle occlusion. As observed, the model maintains robust lane reconstruction capabilities even when the visual information is corrupted by extreme weather or significantly obstructed by other vehicles.

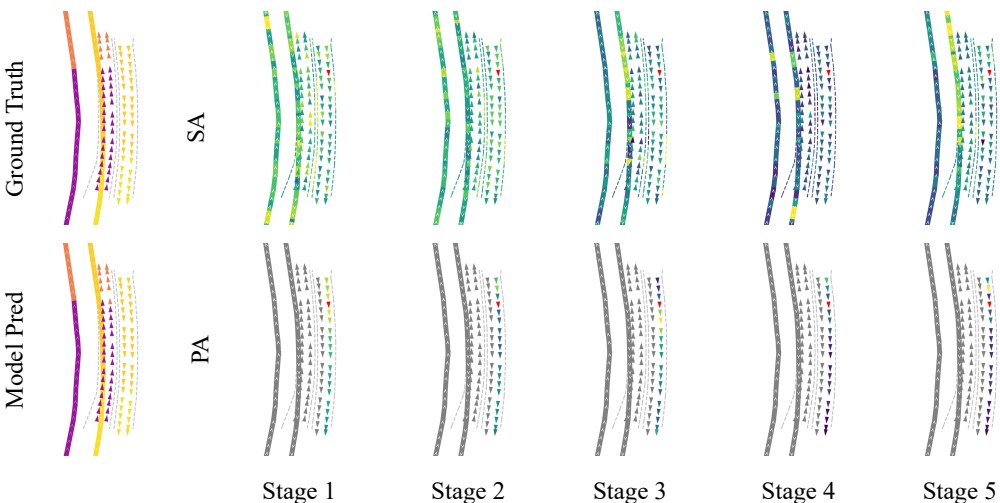

Figure 13: Visualization of attention map of Path-aware attention and spatial attention. SA means Spatial Attention. PA mean Path-aware Attention. The red triangle represents the token corresponding to the current attention map.

our predictions and the ground truth remain, indicating potential for further enhancement. We propose that improving the path-aware attention (PA) mechanism by incorporating local operators such as convolutional kernels could be advantageous. This hybrid approach would preserve model

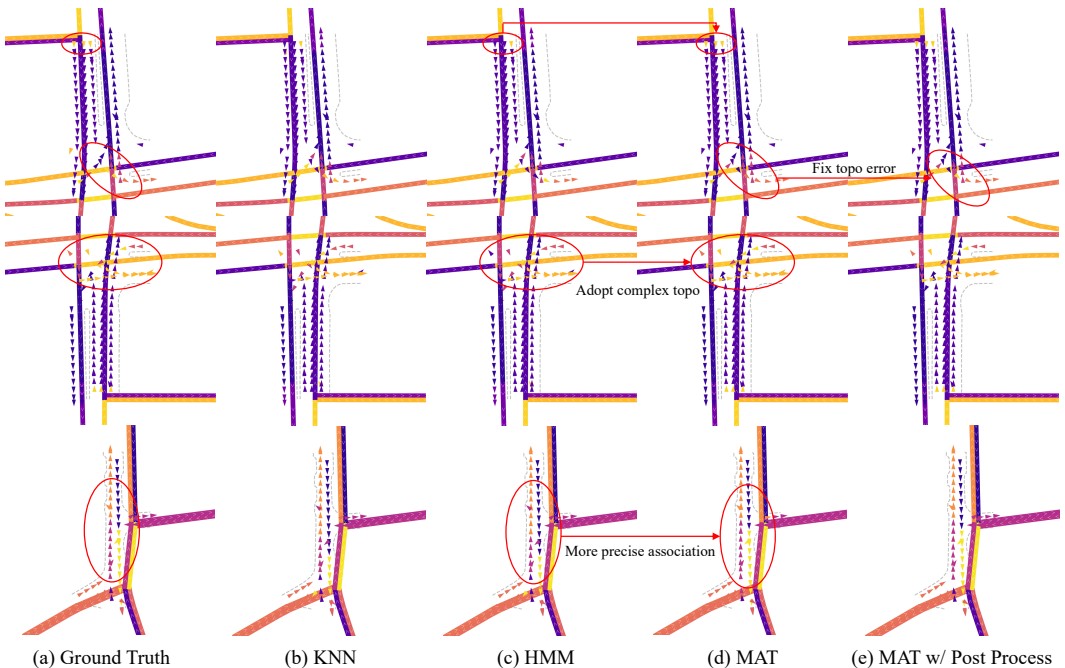

Figure 14: Visualization of result in OMA Val set.

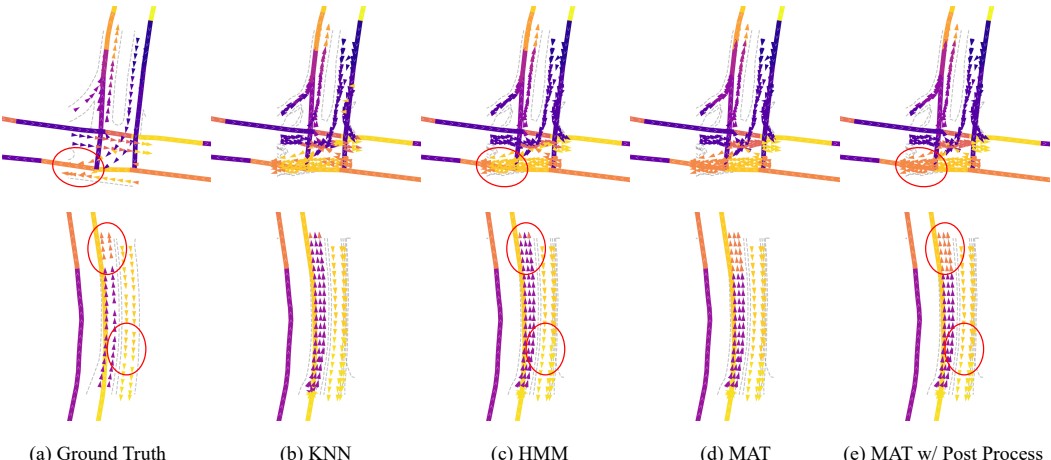

Figure 15: Visualization of result in OMA Test set.

efficiency while allowing for more precise spatial-temporal feature extraction at path intersections, thus improving local association accuracy without compromising inference speed.

### H.4 GENERALIZATION ON DIFFERENT OP GENERATORS

To further validate the robustness of MAT against varying noise patterns inherent to different map generation paradigms, we visualize the association results on the OMA Test set using three distinct baselines: MapTR (Liao et al., 2023a), MapTRv2 (Liao et al., 2023b), and SeqGrowGraph (SSG) (Xie et al., 2025a).

As illustrated in Figure 17, each generator exhibits unique geometric and topological characteristics:

- **Row 1 (MapTR):** Often produces fragmented centerlines with localized geometric jitter, creating a challenge to ensure topological continuity.

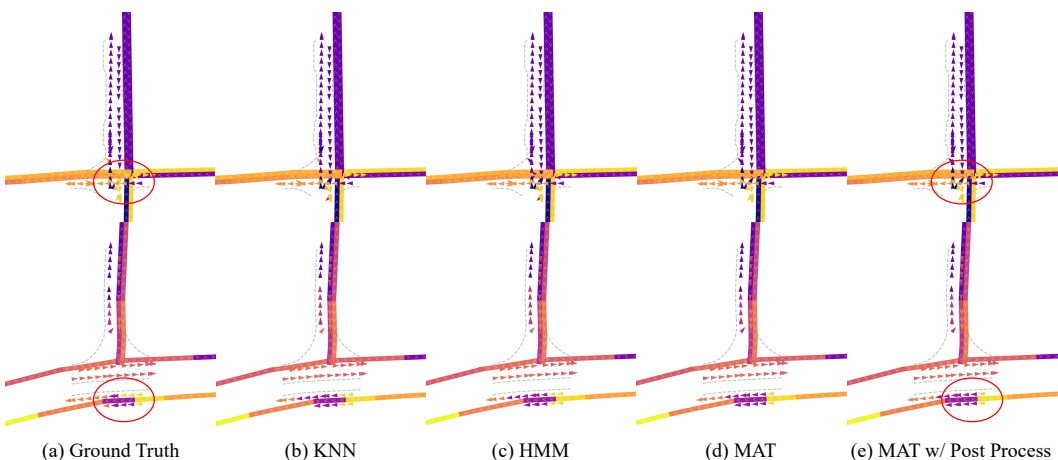

| (a) Ground Truth | (b) KNN | (c) HMM | (d) MAT | (e) MAT w/ Post Process |

Figure 16: Visualization of failed cases.

- **Row 2 (MapTRv2):** Offers improved geometric stability, but still suffers from occasional detection gaps and instance discontinuities.
- **Row 3 (SeqGrowGraph):** Generates highly connected graphs via expansion, which reduces fragmentation, but may introduce erroneous topological links (over-connection) in complex intersections.

Despite these substantial domain gaps, MAT consistently establishes accurate associations (indicated by the consistent coloring of lane instances that match the SD map topology) across all three inputs. This qualitative evidence reinforces the quantitative results in Tab. 1, demonstrating that our proposed Path-Aware and Spatial Attention mechanisms effectively generalize across disparate upstream perception generators without requiring generator-specific fine-tuning.

## I    IMPLEMENT DETAILS

### I.1    MODEL SETTINGS

Tab. 22 summarizes the architectural configurations of our proposed MAT variants (MAT-T and MAT-L). All variants adopt identical channel dimensions , attention head counts, spatial curve orders and hybrid attention mechanisms combining spatial attention (SA) and path-aware attention (PA). In particular, parameters such as patch sizes, MLP ratios (matching spatial curve orders), and stochastic depth rates are uniformly inherited across architectures, reflecting ablation study results that optimized these values for balanced accuracy-latency trade-offs. A distinctive design choice lies in the shuffling strategy, where MAT-T/L progressively refine the shuffle order to enhance token mixing in spatial attention, aligning with their increasing computational budgets. This structured configuration hierarchy enables systematic evaluation of model capacity versus efficiency, as validated by the ascending La / ms metrics ($34 \rightarrow 70$) corresponding to deeper transformer layers.

### I.2    TRAIN CONFIGURATION

The details of the implementation are summarized in Tab. 23. Our training protocol uses the AdamW optimizer with a base learning rate of $1 \times 10^{-4}$ and a cosine learning rate decay, operating in batch size of 128. The weight decay regularization is set to $5 \times 10^{-3}$. The training process spans 50 epochs with a 2-epoch warm-up phase for learning rate initialization. Model optimization combines CrossEntropy loss for classification tasks and CTC loss for sequence alignment objectives.

### I.3    DATA AUGMENTATIONS

For data augmentation as shown in Tab. 23, we implement a series of randomized transformations that include axis-aligned rotation around the z-axis with $\pm 1°$ angular variation at a 50% application

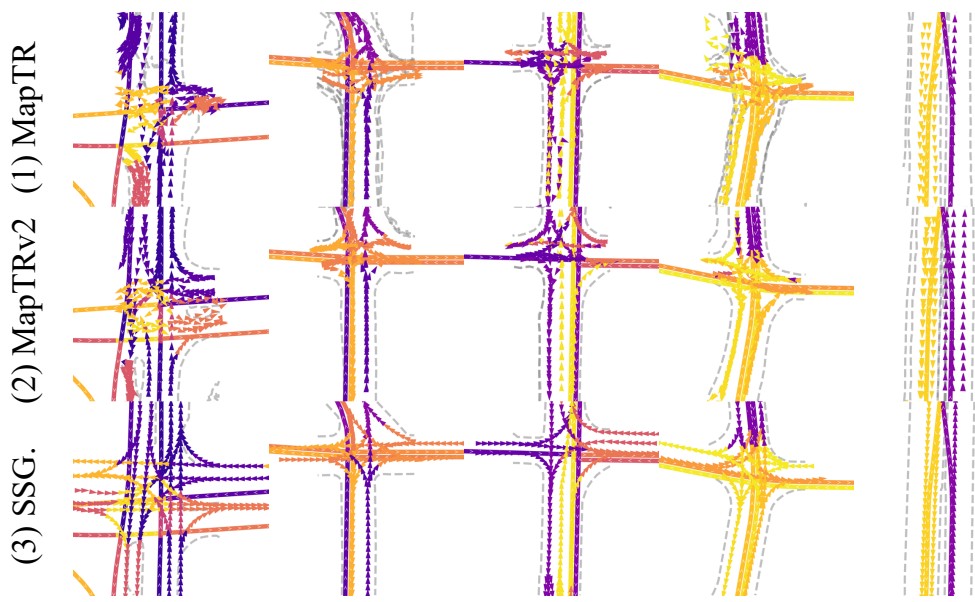

Figure 17: Visualization of MAT association results across different Online Perception (OP) map generators on the OMA Test set. Rows correspond to (1) MapTR, (2) MapTRv2, and (3) Seq-GrowGraph (SSG). The columns display diverse scenarios including intersections and straight roads. Consistent coloring between lane segments indicates that MAT successfully associates fragmented or noisy OP elements with the correct semantic road instances from the SD map, demonstrating strong cross-generator robustness.

Table 22: Model settings.

| Parameter | MAT-T | MAT-L |
|---|---|---|
| Blocks | [2, 2, 2, 2, 2] | [4, 4, 4, 12, 4] |
| Attention Head | | [4, 4, 8, 8, 8] |
| MLP Ratio | | [4, 4, 4, 4, 4] |
| Drop Path | | [0.3, 0.3, 0.3, 0.3, 0.3] |
| Channels | | [96, 192, 384, 768, 1536] |
| Path Size | | [1024, 1024, 1024, 1024, 1024] |
| Attention Order | | ["Spatial Attention", "Path-aware Attention"] |
| Spatial Curve | | ["z", "z-trans", "hilbert", "hilbert-trans"] |
| Shuffle | | [Shuffle Order, Shuffle Order, Shuffle Order, Shuffle Order, Shuffle Order] |
| Latency/ms | 34 | 70 |

Table 23: Train Configuration and Data augmentations.

| Training Configuration | | | |
|---|---|---|---|
| optimizer | AdamW | batch size | 128 |
| scheduler | Cosine | weight decay | 5e-3 |
| learning rate | 1e-4 | epochs | 50 |
| criteria | CrossEntropy, CTC Loss | warmup epochs | 2 |
| Data Augmentation | | | |
| random rotate | axis: z, angle: [-1, 1], p: 0.5 | random scale | scale: [0.9, 1.1] |
| random flip | p: 0.5 | random jitter | sigma: 0.005, clip: 0.02 |
| grid sampling | grid size: [0.1, 0.1, $\pi/16$] | | |

probability, isotropic scaling within the range $[0.9, 1.1]$, random flipping with equal probability

50%, point cloud jittering characterized by $\sigma = 0.005$ and a clip limit of 0.02, and grid sampling with spatial discretization parameters set to $[0.1, 0.1, \pi/16]$. These enhancement strategies were systematically validated through ablation studies to optimize the balance between model accuracy and computational efficiency while ensuring robustness to input variations.

