# OpenReview forum: "Online Navigation Refinement: Achieving Lane-Level Guidance by Associating Standard-Definition and Online Perception Maps"
_ICLR.cc/2026/Conference — ICLR 2026 Poster_

### Official Review · Reviewer_FhYL · 2025-10-31

**Soundness:** 3
**Presentation:** 3
**Contribution:** 4
**Rating:** 8
**Confidence:** 4

**Summary:**

This paper proposes a new and important task for Online Perception Maps and SD map association. It first introduces Navigation Refinement P-R to evaluate the geometric and association accuracy, and then proposes a model named MAT to take centerlines, SDMap roads and boundary lines as input to predict road-lane association.
In the MAT model, two modules - PA and SA are carefully designed to consider topological and spatial features learning.
The paper also proposes the online map association dataset (OMA). Overall, this work focuses on a new problem of map association for low-cost and up-to-date lane-level navigation.

**Strengths:**

1. This paper proposes a new and important task for the online perception maps association, which is valuable for real-world applications in autonomous driving. Assigning prediction lanes to SD road elements is essential for building topological relationships between lanes and roads, further benefiting navigation and planning. Building a corresponding benchmark and evaluation metric fills the gap in the field of online map construction, as far as I know.
2. The evaluation metrics are carefully designed to cover various cases, which reflect the distance matching, connection assignment, and length accuracy ratio.
3. The contrastive experiments, ablation study and analysis are thorough, which demonstrates the effectiveness of each part of the model and post-process algorithm.

**Weaknesses:**

1. For the subfigure Fig 2.(3), there are two junction curve lines connecting two crossroads. The mapping of such transition
 lines is ambiguous due to their connection attributes. How to solve this problem when constructing the ground truth assignment labels?
2. As this paper also studies the noise of road elements and assignment with predicted lanes, a line of related works regarding the use of SDMaps and the noise problem for map perception should be included and discussed, such as SMERF [1],  TopoSD [2], P-MapNet [3] and etc.
3. We know there are inevitably some cases with the absence of the SDMap road elements or shifting road geometry,  caused by the errors of GPS localization or map construction.  We would like to know how these factors affect the association process and how the proposed metrics maintain robustness when dealing with imperfect SDMaps. In particular, this concern applies to cases where the generated online maps are geometrically or topologically inaccurate, even though they are otherwise precise.

[1] Luo K Z, Weng X, Wang Y, et al. Augmenting lane perception and topology understanding with standard definition navigation maps[C]//2024 IEEE International Conference on Robotics and Automation (ICRA). IEEE, 2024: 4029-4035.
[2] Yang S, Jiang M, Fan Z, et al. Toposd: Topology-enhanced lane segment perception with sdmap prior[J]. arXiv preprint arXiv:2411.14751, 2024.
[3] Jiang Z, Zhu Z, Li P, et al. P-MapNet: Far-seeing Map Generator Enhanced by both SDMap and HDMap Priors[J]. arXiv preprint arXiv:2403.10521, 2024.

**Questions:**

1. In Figure 4, the caption states that “Label overlap in both paths is 67%. TP for T=50% but FP for T=95%.” I am wondering how the value of Label overlap is computed — is it defined as the ratio of correctly matched (or overlapping) edges to the total number of edges? How is an edge seen as the correct one?
2. Why does the order of road tokens or centerline tokens matter in the MAT attention modules?

---

> ### Author Response · Authors · 2025-11-24
> **Comment for Review FhYL - 1**
>
> We are grateful for the Reviewer's **valuable and helpful comments**. We highly appreciate the time spent on reviewing our work and enhancing the quality of our submission.
>
> Please find our point-by-point response below. We have also marked the corresponding changes in the updated paper to facilitate your review. We believe these revisions have significantly strengthened our work.
>
> ---
>
> **Weaknesses 1**:
>
> For the subfigure Fig 2.(3), there are two junction curve lines connecting two crossroads. The mapping of such transition lines is ambiguous due to their connection attributes. How to solve this problem when constructing the ground truth assignment labels?
>
> **Response for Weaknesses 1**:
>
> We sincerely thank the Reviewer for the detailed reading and the keen observation regarding the transition lines in Figure 2~(3). We acknowledge that the assignment of road in such transition areas indeed presents inherent ambiguity due to the heterogeneity between SD Maps and OP Maps. To mitigate the potential impact of this ambiguity on both training stability and evaluation fairness, we have designed specific strategies for both the annotation process and the evaluation metrics:
>
> * **Standardized Annotation Protocol**: To resolve ambiguity during ground truth construction, we established a unified rule: the assignment of a Lane Vector associated to an road in SD map is determined by **the the closest vector of road to the start point of that lane vector**. Furthermore, we ensure that the topological changes in the Lane Vector associations remain consistent with the transitions of the SD Links.
>
> * **Tolerance in Evaluation Metrics**: To minimize the evaluation error caused by boundary ambiguity, our metric assesses the path-level association accuracy. We evaluate performance across 11 thresholds ranging from 50% to 95%. Crucially, even at the strictest threshold (95%), the remaining 5% tolerance buffer can accommodate the inevitable semantic ambiguity at the transition boundaries of SD Links.
>
> **Paper Enhancement**: To further clarify this ambiguity, we have added a **detailed discussion regarding the dataset annotation rules in the Appendix**. Additionally, we have revised Figure 2~(3) to ensure the schematic representation more accurately reflects the actual ground truth assignment logic.

---

> > ### Author Response · Authors · 2025-11-24
> > **Comment for Review FhYL - 2**
> >
> > **Weaknesses 2**:
> >
> > As this paper also studies the noise of road elements and assignment with predicted lanes, a line of related works regarding the use of SDMaps and the noise problem for map perception should be included and discussed, such as SMERF [1], TopoSD [2], P-MapNet [3] and etc.
> >
> > **Response for Weaknesses 2**:
> >
> > We sincerely appreciate this constructive suggestion. We agree that discussing these works strengthens the context of our study regarding SD map utilization and noise resilience. We have incorporated the recommended citations  into the Related Work of the revised manuscript. We have added a discussion on how these methods utilize SD map priors to address perception noise, positioning our contribution alongside them.

---

> ### Author Response · Authors · 2025-11-24
> **Comment for Review FhYL - 3**
>
> **Weaknesses 3**:
>
> We know there are inevitably some cases with the absence of the SDMap road elements or shifting road geometry, caused by the errors of GPS localization or map construction. We would like to know how these factors affect the association process and how the proposed metrics maintain robustness when dealing with imperfect SDMaps. In particular, this concern applies to cases where the generated online maps are geometrically or topologically inaccurate, even though they are otherwise precise.
>
> **Response for Weaknesses 3**:
>
> Thank you for your insightful comments regarding the impact of imperfect SD Maps, such as GPS errors, geometric offsets, and missing elements, on the association process. We fully agree that robustness to such noise is critical for practical applications.
>
> To address your concerns, we first clarify the robustness of our evaluation metrics. **The NR-F1 metric calculates performance by comparing the generated results against the annotated HD Map ground truth**, which is independent of the SD Map input. Therefore, the metric faithfully reflects how input noise impacts model performance without bias.
>
> To rigorously quantify the robustness of our proposed method, **we conducted additional ablation studies on the OMA**. We simulated three types of SD Map imperfections by injecting varying levels of noise (Low, Medium, High):
>
> * **Global Shift**: Simulates GPS localization errors (tested at 10%, 20%, and 50% offset ratios of OP map range).
> * **Element Noise**: Simulates geometric construction errors (tested at 5%, 10%, and 20% jitter ratios of OP map range).
> * **Element Absence**: Simulates topological incompleteness (tested at 10%, 20%, and 30% omission ratios of OP map range).
>
> The results (NR-F1 score) are summarized below:
>
> **Tab R3-1: Ablation study for robustness of SD map noise.**
> | Noise Type | Low Noise (Val/Test) | Medium Noise (Val/Test) | High Noise (Val/Test) |
> | :--- | :---: | :---: | :---: |
> | **Global Shift** | 77.5 / 43.3 | 76.0 / 42.9 | 74.6 / 41.4 |
> | **Element Noise** | 78.3 / 44.8 | 77.5 / 44.3 | 77.5 / 44.1 |
> | **Element Absence** | 77.5 / 44.0 | 76.4 / 43.7 | 75.5 / 43.1 |
>
> **It is crucial to note that the Test set inherently contains real-world noise**, including topological errors and spatial offsets, which aligns with the concerns raised. The results above show that even when additional synthetic noise is superimposed onto these existing imperfections, **the performance remains within an acceptable range**. This confirms our method's resilience in two key aspects:
>
> 1. **Resilience to Cumulative Error**: In the Test set, despite the presence of inherent data noise plus high levels of injected Global Shift (50%), the model still retains reasonable effectiveness (41.4 in test set). This demonstrates that our Path-Aware Attention successfully captures the relative topological structure, reducing dependency on absolute coordinate alignment.
>
> 2. **Stability against Topology & Geometry Changes**: For Element Noise and Absence, the performance drop on the Test set is marginal (e.g., from 44.8 to 44.1). This indicates that our Spatial Attention effectively aggregates global context, ensuring the system does not fail even when dealing with the "double challenge" of real-world errors and artificial perturbations.
>
> **Paper Enhancement:** To comprehensively address this concern, we have added a dedicated section in the **Appendix** detailing the ablation studies on model robustness against SD Map noise.

---

> ### Author Response · Authors · 2025-11-24
> **Comment for Review FhYL - 4**
>
> **Question 1**:
>
> In Figure 4, the caption states that “Label overlap in both paths is 67%. TP for T=50% but FP for T=95%.” I am wondering how the value of Label overlap is computed — is it defined as the ratio of correctly matched (or overlapping) edges to the total number of edges? How is an edge seen as the correct one?
>
> **Response to Question 1**:
>
> Thank you for this insightful question. You are correct that Label Overlap is defined as the ratio of the length of correctly matched segments (where the predicted road ID matches the Ground Truth road ID) to the total path length.
>
> To explicitly answer "how an edge is seen as the correct one" and how the computation works, the process follows a strict three-stage protocol outlined in Section 4.2:
>
> * **1. Geometric Alignment (Stage 1):** First, we establish a candidate correspondence between a predicted path and a Ground Truth (GT) path using the bidirectional Chamfer distance. If a predicted path is geometrically too far from any GT path (exceeding the distance threshold, e.g., 1 meter), it is immediately classified as a False Positive (FP).
>
> * **2. Label Sequence Alignment Verification (Stage 2):** Before calculating the specific overlap percentage, we verify the topological correctness. We convert the geometrically matched paths into simplified sequences of road IDs (e.g., Road A $\to$ Road B). If the predicted sequence of road IDs does not align with the GT sequence (e.g., incorrect order or mismatching road IDs), the prediction is marked as an FP. This step ensures that the path is not just geometrically close, but topologically valid.
>
> * **3. Length-Aware Overlap Assessment (Stage 3):** For paths that pass the sequence verification, we then calculate the Label Overlap. We measure the length of the segments where the predicted road ID is identical to the GT road ID. This length is then divided by the total length of the GT path. If this ratio exceeds the intersection-over-union (IoU) threshold T (e.g., T=50%), the prediction is finally classified as a True Positive (TP).
>
> **Paper Enhencement**: We have added a clear definition of Label Overlap in Section 4.2.

---

> > ### Author Response · Authors · 2025-11-24
> > **Comment for Review FhYL - 5**
> >
> > **Question 2**:
> >
> > Why does the order of road tokens or centerline tokens matter in the MAT attention modules?
> >
> > **Resonpse for Question 2**:
> >
> > thank you for this insightful question. The strict ordering of tokens is a fundamental requirement derived from our architecture design. We break this down into three key aspects:
> >
> > 1. **Computational Constraints of Group Attention**:
> >
> > To achieve the real-time inference, MAT employs Group Attention (also named as Shifted Window Self-Attention in Swin Transformer). This reduces the computational complexity from the standard global O(N^2) to linear O(N) but imposes a hard constraint: the receptive field is strictly limited to a local window (the group). Consequently, the token order directly dictates the neighborhood. With random ordering, the attention mechanism would operate on unrelated, fragmented nodes, failing to capture meaningful context.
> >
> > 2. **Ordering as an Inductive Bias for Grouping**:
> >
> > To maximize feature interaction within these limited receptive fields, we must introduce structural priors. We treat token ordering as a clustering strategy: it forces highly correlated tokens to be adjacent in the sequence. When the sequence is sliced into groups, these related tokens naturally fall into the same attention bucket. This ensures the attention heads focus their limited computational budget on high-value correlations rather than noise.
> >
> > 3. **Modeling Orthogonal Map Properties**:
> >
> > Road networks possess two distinct, "orthogonal" attributes: geometry and topology. MAT explicitly models both by strictly enforcing two types of ordering:
> > * **Spatial Order: Encodes geometric proximity**. This ensures physically adjacent entities—even if topologically distant or belonging to different map layers (SD vs. OP)—interact within the same group.
> > * **Topological Order: Encodes connectivity**. This ensures that predecessor and successor nodes interact within the same group.
> >
> > **Paper Enhancement**: To address this concern, we have refined the Sec. 5.2 and Sec 5.3 of Path-Aware Attention and Spatial Attention, incorporating a more detailed discussion on their design motivation and theoretical analysis.

---

### Official Review · Reviewer_9aX1 · 2025-11-01

**Soundness:** 3
**Presentation:** 3
**Contribution:** 3
**Rating:** 6
**Confidence:** 3

**Summary:**

Targeting the pain points regarding cost and real-time performance in lane-level navigation, this paper proposes a new task of Online Navigation Refinement (ONR). By associating Standard-Definition Maps (SD) with Online Perception Maps (OP), it upgrades road-level navigation to lane-level navigation, while simultaneously addressing the core issues that existing solutions rely on expensive HD maps and OP maps lack global topology.

**Strengths:**

1. It directly tackles the core pain points of existing lane-level navigation solutions. The proposed ONR task bridges SD and OP maps to achieve low-cost, up-to-date lane-level guidance, which is highly relevant to real-world needs in GIS and autonomous driving.
2. The paper provides a "dataset-model-metric" trinity solution to fill research gaps:OMA Dataset, MAT Model and NR P-R Metric.
3. The paper provided solid experimental validation.

**Weaknesses:**

1. The test set's OP map noise only comes from MapTRv2.  It does not evaluate MAT’s performance under other common OP noise types (e.g., severe lane occlusion by vehicles, sensor failure in heavy rain/fog), I think it should be evaluated to improve the model's robustness.
2. The test set uses OP maps generated solely by MapTRv2. If other OP map generators produce different noise patterns, whether MAT’s performance will degrade or not remains untested, i think the cross-generator generalization ability should be evaluated.

**Questions:**

1. The OMA dataset’s annotations are manually completed. Is there a plan to introduce semi-supervised/unsupervised annotation tools to reduce costs for future dataset expansion?
2. The metric aggregates results across 15 length intervals to avoid short-path bias. However, it does not consider road complexity (e.g., straight roads and curved roads). Could we adding a "complexity weight" to improve the metric’s fairness in evaluating model performance on diverse road types?

---

> ### Author Response · Authors · 2025-11-24
> **Comment for Reviewer 9aX1 - 1**
>
> We are grateful for your **detailed and constructive feedback**. We fully acknowledge the validity of your points and believe that addressing them has significantly strengthened our paper.
>
> Please find our point-by-point response below. We have also marked the corresponding changes in the updated paper to facilitate your review. We believe these revisions have significantly strengthened our work.
>
> ---
>
> ### Response for Weaknesses 1:
>
> We thank you for the constructive suggestion. We fully agree that evaluating the model under diverse noise conditions, such as occlusion and severe weather, is critical for verifying robustness.
>
> To address this, we have conducted extended experiments focusing on two aspects: quantitative evaluation on corrupted inputs (environmental noise) and qualitative analysis of occlusion scenarios. These results are detailed in **Appendix G.3** of the revised paper.
>
> **1. Clarification on "Severe Lane Occlusion"**
>
> While explicit quantitative evaluation is constrained by the lack of ground-truth occlusion annotations in public datasets, we analyze the occlusion problem by visualization.
>
> * **Qualitative Visualization:** We present a case study of **severe vehicle occlusion in Figure 12** in Appendix. As shown in the visualization, the upstream perception model (MapTRv2) manages to reconstruct a coherent global road structure despite significant visual blockage by vehicles, thanks to its temporal modeling design. Taking advantage of this inherent robustness, our downstream MAT model successfully associates the lanes with the correct SD map topology, ensuring stable navigation refinement even under dynamic occlusion.
>
> **2. Extended Evaluation on Environmental Noise (Rain, Fog, and Dark)**
>
> To rigorously evaluate the model's performance under "sensor failure" caused by environmental factors, we conducted a comprehensive experiment using **nuScenes-C** [r1], a benchmark specifically designed for assessing robustness against input corruptions.
>
> * **Experimental Setup:** We utilized the MapTRv2 model trained on clean nuScenes data)to perform inference on the nuScenes-C validation set. This generates OP maps reflecting realistic sensor degradation scenarios under three specific weather conditions: **Rain**, **Fog**, and **Dark (Night)**.
> * **Zero-Shot Inference:** We then applied our MAT model to associate these noisy OP maps with SD maps. **Crucially, we did not retrain MAT on these corrupted samples.** This zero-shot setting serves as a rigorous stress test for the model's generalizability.
>
> **Quantitative Results:**
> The results (Metric: $NR-F1^{50:95}$) across three difficulty levels (Easy, Mid, Hard) are reported in **Table 21**:
>
> | Condition | Easy | Mid | Hard |
> | :--- | :---: | :---: | :---: |
> | **Rain** | 44.7 | 42.1 | 40.6 |
> | **Fog** | 44.2 | 43.4 | 42.1 |
> | **Dark** | 44.0 | 43.8 | 41.6 |
>
> **Analysis:**
> As shown in the table, MAT maintains robust performance across all conditions. Even under "Hard" settings where visual features are significantly compromised, the performance drop is minimal. For instance, in Fog scenarios, the metric only decreases from 44.2 (Easy) to 42.1 (Hard). This stability demonstrates that by leveraging the topological priors from SD maps, MAT effectively mitigates the impact of sensor noise.
>
> **Qualitative Visualization:**
> We further illustrate these scenarios in **Figure 10, 11, 12** in appendix of page 29, 30 and 31:
> * **Figure 10** displays the synthesized degradation samples (Rain, Fog, Dark) across varying difficulty levels.
> * **Figure 11, 12** compares the inference results under the "Hard" setting. Despite severe noise in the input OP maps (e.g., missing boundaries or phantom lanes due to fog/rain), MAT successfully recovers the correct lane topology by aligning it with the SD map.
>
> **Paper Enhancement**: Following your suggestion, we have added a subsection G.3 to the ablation studies in the Appendix G, along with relevant tables and figures.
>
> [r1] Xie, Shaoyuan, et al. "Benchmarking and improving bird's eye view perception robustness in autonomous driving." IEEE Transactions on Pattern Analysis and Machine Intelligence (2025).

---

> ### Author Response · Authors · 2025-11-24
> **Comment for Reviewer 9aX1 - 2**
>
> **Weakness 2**:
>
> The test set uses OP maps generated solely by MapTRv2. If other OP map generators produce different noise patterns, whether MAT’s performance will degrade or not remains untested, i think the cross-generator generalization ability should be evaluated.
>
> **Response for Weaknesses 2**:
>
> Thank you for this valuable suggestion regarding the generalization capability of our model across different OP map generators. We agree that verifying robustness against varying noise patterns is crucial for real-world deployment.
>
> To address this, we have extended our evaluation on the OMA Test set by incorporating OP maps generated by two additional methods: MapTR [r1] and SeqGrowGraph [r2]. The MapTR is trained by a config with centerline, similar as MapTRv2. In detail, we evaluated the NR-F1$^{50:95}$ metric across all baselines and our proposed MAT models. The results are summarized in the table below:
>
> **Tab R2-2: Result on OMA Test set. La. means latency. MM, PM, GM, MA means map matching, graph matching, point matching and map association method. P-M and M-M means the model is trained and inferred by path-to-map or map-to-map. SGG. means SeqGrowGrpah.**
>
> | Methods | Present | Type | Paradigm | MapTR (NR-F1) | MapTRv2 (NR-F1) | SGG. (NR-F1) | La./ms |
> | :--- | :---: | :---: | :---: | :---: | :---: | :---: | :---: |
> | HMM | SIGSPATIAL'09 | MM | P-M | 33.1 | 36.0 | 46.5 | 561 |
> | DeepMM | TMC'20 | MM | P-M | 32.8 | 33.5 | 43.9 | 733 |
> | MTrajRec | KDD'21 | MM | P-M | 34.1 | 36.8 | 48.9 | 1593 |
> | GraphMM | TKDE'23 | MM | P-M | 35.9 | 38.7 | 49.2 | 889 |
> | EAM³ | TITS'25 | MM | P-M | 36.3 | 39.1 | 50.8 | 679 |
> | KNN | -- | GM | M-M | 31.7 | 34.6 | 43.5 | 313 |
> | GMT | TPAMI'24 | GM | M-M | 30.1 | 33.4 | 42.3 | 105 |
> | FastMAC | CVPR'24 | PM | M-M | 32.0 | 35.8 | 45.0 | 81 |
> | MAT-T (Ours) | -- | MA | M-M | 41.5 | 44.8 | 54.8 | **35** |
> | MAT-L (Ours) | -- | MA | M-M | **41.9** | **45.0** | **54.9** | 74 |
>
> According to the tab, we can find that:
> * **Consistent Superiority**: Our method (MAT) consistently outperforms all state-of-the-art baselines on both new map generators. On the SeqGrowGraph dataset, MAT-L achieves an NR-F1$^{50:95}$ of 54.9%, surpassing the strongest baseline (EAM$^3$) by 4.1%. On the MapTR dataset, which presents more challenging noise patterns (reflected in lower scores across all methods), MAT-L maintains a significant lead of 5.6% over EAM$^3$.
> * **Robustness to Noise Patterns**: Different generators produce distinct topological errors (e.g., MapTR often produces fragmented lane segments, while graph-growing methods may introduce incorrect connectivity). The results demonstrate that MAT's architecture effectively generalizes across these disparate noise distributions without requiring specific tuning for the generator.
> * **Generalizability of Models and Metrics**: In graph construction, SeqGrowGraph significantly outperforms maptr and maptrv2. In the final evaluation results, our model also demonstrates substantial self-improvement, reflecting the generalizability of both the model methodology and corresponding metrics.
>
> Furthermore, we have provided Figure 17 on the page 34, a visualization of MAT association results across different Online Perception (OP) map generators. Consistent coloring between lane segments indicates that MAT successfully associates fragmented or noisy OP elements with the correct semantic road instances from the SD map, demonstrating strong cross-generator robustness.
>
> **Paper Enhancement**: We have included these additional results in the revised paper and update the Sec 6.2 to further validate the cross-generator generalization ability of the proposed Online Navigation Refinement framework.
>
> [r1] Liao, Bencheng, et al. "MapTR: Structured Modeling and Learning for Online Vectorized HD Map Construction." The Eleventh International Conference on Learning Representations.
>
> [r2] Xie, Mengwei, et al. "Seqgrowgraph: Learning lane topology as a chain of graph expansions." Proceedings of the IEEE/CVF International Conference on Computer Vision. 2025.

---

> ### Author Response · Authors · 2025-11-24
> **Comment for Reviewer 9aX1 - 3**
>
> **Question 1**:
>
> The OMA dataset’s annotations are manually completed. Is there a plan to introduce semi-supervised/unsupervised annotation tools to reduce costs for future dataset expansion?
>
> **Response for Question 1**:
>
> We appreciate the reviewer’s forward-looking suggestion. We indeed plan to expand our benchmark to include other datasets (e.g., Argoverse) in the future. To minimize annotation costs during this expansion, we will adopt a "Human-in-the-Loop" iterative annotation strategy). Specifically, our pipeline for expanding to new datasets is as follows:
>
> 1. **Model-Assisted Pre-annotation**: We will utilize the current MAT model trained on the OMA (nuScenes) dataset to perform zero-shot or few-shot inference on the unlabeled new data. This generates initial "draft" associations (e.g., SD-to-HD correspondences and topology).
>
> 2. **Lightweight Manual Correction**: Annotators will focus solely on verifying and correcting the model's predictions rather than labeling from scratch. This significantly reduces the workload.
>
> 3. **Closed-Loop Iteration**: The corrected data is then added to the training set to fine-tune the model, which is subsequently used to pre-annotate the next batch of data with improved accuracy.
>
> To validate the feasibility of this strategy, we conducted a **Data Efficiency experiment** on the OMA dataset. We trained our model using varying percentages of the training data and evaluated performance on the full Validation and Test sets. The results are shown below:
>
> **Tab R2-3: Data Efficiency experiment of MAT-L on OMA**
>
> | Metric | 1% | 2% | 5% | 10% | 20% | 50% | 100% |
> | :--- | :---: | :---: | :---: | :---: | :---: | :---: | :---: |
> | **Val / NR-F1^${50:95}$** | 24.9 | 64.4 | 77.1 | 77.7 | 77.3 | 78.3 | 78.7 |
> | **Test / NR-F1^${50:95}$** | 22.3 | 41.5 | 44.5 | 44.6 | 46.6 | 44.7 | 45.0 |
>
> The experimental results demonstrate exceptional data efficiency. With **only 5%** of the annotated training data, the model achieves an NR-F1 of 77.1% on the validation set and 44.5% on the test set, which is comparable to the performance using 100% of the data (78.7% / 45.0%).
>
> This finding strongly supports our proposed pipeline that since the model can learn robust association features with very little data, it can provide high-quality pre-annotations early in the expansion process, thereby drastically reducing manual effort.
>
> **Paper Enhancement**:  We have added the analysis and experiment on the Section 6.3 and Appendix D.7.

---

> ### Author Response · Authors · 2025-11-24
> **Comment for Reviewer 9aX1 - 4**
>
> **Question 2**:
>
> The metric aggregates results across 15 length intervals to avoid short-path bias. However, it does not consider road complexity (e.g., straight roads and curved roads). Could we adding a "complexity weight" to improve the metric’s fairness in evaluating model performance on diverse road types?
>
> **Response for Question 2**:
>
> We thank you for this insightful suggestion. We agree that road complexity is a critical factor affecting map association performance, and the curvature of the road may also be a significant factor affecting the association.
>
> To address this, we designed a Curvature-based Weighted Metric to evaluate the discrepancy between straight and curved paths. Specifically:
>
> * **Complexity Measure**: Since calculating curvature via average angles is highly sensitive to point sampling rates, we employed the Discrete Fréchet Distance between the actual path and the straight line connecting its endpoints as a robust measure of geometric complexity.
>
> * **Stratification**: Similar to our length-based approach, we discretized the complexity scores into 10 bins. The reason we continue to use stratified statistics is that we have statistically analyzed the overall complexity and found that, like length, it still exhibits a pronounced long-tail distribution, **as shown in the Figure 8 on page 23**. So we selected the 95th percentile of complexity across all paths as an upper bound and uniformly divided this upper bound into 10 bins. We computed the NR-F1 score for each bin and also calculated a weighted average (where each complexity level contributes equally) to assess fairness.
>
> The result of the NR-F1 of MAT in OMA under the Curvature-based Weighted Metric is shown in the Tab.R2-4 and Tab.R2-5:
>
> **Table R2-4: Impact of Complexity Stratification**
>
> | Complexity Bins (N) | 1 (Unweighted) | 2 | 5 | 10 (Fully Weighted) |
> | :--- | :---: | :---: | :---: | :---: |
> | **Val** | 81.6 | 70.2 | 62.7 | 60.3 |
> | **Test** | 51.8 | 38.7 | 33.5 | 32.5 |
>
> **Table R2-5: Performance per Complexity Bin**
>
> | Bin Index | 1 | 2 | 3 | 4 | 5 | 6 | 7 | 8 | 9 | 10 |
> | :--- | :---: | :---: | :---: | :---: | :---: | :---: | :---: | :---: | :---: | :---: |
> | **Val** | 87.2 | 63.2 | 56.7 | 50.7 | 49.9 | 49.1 | 46.5 | 45.2 | 42.7 | 43.1 |
> | **Test** | 57.9 | 50.4 | 32.0 | 21.2 | 16.8 | 147. | 10.7 | 10.6 | 10.4 | 11.6 |
>
> According to the Tab.R2-4, as we increase the number of bins (forcing the metric to weigh complex roads equally to straight ones), the overall score drops significantly. This confirms that the dataset is dominated by simple roads where the model performs well.
>
> Furthermore, the statistics on Tab.R2-5 reveal a long-tail distribution similar to the path length distribution: the vast majority of roads are straight (Bin 1), where the model achieves high accuracy (Val: 0.872). However, performance drops drastically on high-Fréchet-distance paths (Bin 10 Val: 0.431), indicating a strong correlation between road complexity and association difficulty.
>
> Finnaly, we believe this metric, along with length-based metrics, **reflects one aspect of road complexity assessment**. We have added this analysis to the Appendix E.2 to encourage future research in road complexity evaluation.
>
> **Paper Enhancement**: We have added this analysis to the Appendix E.2.

---

### Official Review · Reviewer_cbHh · 2025-11-04

**Soundness:** 2
**Presentation:** 2
**Contribution:** 2
**Rating:** 4
**Confidence:** 4

**Summary:**

This work focuses on the task of Online Navigation Refinement, which aims to transform the road-level navigation derived from SD maps into precise lane-level navigation aligned with the online perception maps. Specifically, the authors introduce the Online Map Association Dataset (OMA), which is developed from nuScenes, and a corresponding transformer-based model for the real-time map association task. In addition, to measure the alignment of the paths and the precision of correspondence, the authors also proposed a new metric named Navigation Refinement P-R (NR P-R). MAT is verified on the OMA dataset and achieved rather low latency (34ms) while maintaining comparable performance improvements compared to previous methods.

**Strengths:**

- Although existing autonomous driving datasets, such as nuScenes and OpenLane-V2, provide fine-grained local lane geometry annotations, and OpenStreetMap provides road-level SD Map annotations, the mappings between them are less explored. The OMA dataset proposed by this work greatly mitigates this data gap.
- Path-aware attention is introduced to align the topologies under the distractions introduced by spatial fluctuations and semantic disparities. By forcing each token can appear only once in the path, cycles are prevented from happening and interactions are ensured to occur only between tokens within the same path.
- The authors also proposed a Spatial Attention module that enhances feature interactions at the instance level across a wider spatial scope.

**Weaknesses:**

- Figure 5 is not self-contained. The concepts of path-aware and spatial attention are hard to grasp from the visual alone, as the figure lacks sufficient illustrative descriptions or a detailed caption.
- Abbreviations are sometimes unclear or non-standard. For example, "Para." in Table 1 is ambiguous and should be explicitly defined (e.g., as "Parameters").
- The section on "attention with vector serialization" is underdeveloped. The explanation is too brief, lacking the detail needed for the reader to fully understand the proposed mechanism.

**Questions:**

- What’s the difference between MAT-T and MAT-L?

---

> ### Author Response · Authors · 2025-11-24
> **Comment for Review cbHh - 1**
>
> We thank the Reviewer for **the insightful and constructive feedback**. We have carefully considered all comments and revised our manuscript accordingly.
>
> Please find our point-by-point response below. We have also marked the corresponding changes in the updated paper to facilitate your review. We believe these revisions have significantly strengthened our work.
>
> ---
>
> **Weaknesses 1**:
>
> Figure 5 is not self-contained. The concepts of path-aware and spatial attention are hard to grasp from the visual alone, as the figure lacks sufficient illustrative descriptions or a detailed caption.
>
> **Response for Weaknesses 1**:
>
> We thank you for the feedback on the readability of our figures. We agree that the previous version of Figure 5 lacked sufficient detail to be fully self-contained. To address your concern and clarify the concepts of Path-aware and Spatial Attention, we have revised Figure 5 in the updated manuscript with the following two improvements:
>
> 1. **Enhanced Visualization of Spatial Attention**: We have modified the illustrative diagram of the Spatial Attention module within Figure 5. It now explicitly visualizes the relationship between the spatial filling curve and the serialized token order, intuitively demonstrating how the model utilizes spatial curves to preserve geometric locality during token sorting.
>
> 2. **Expanded Caption**: We have significantly expanded the caption of Figure 5 to provide a self-contained explanation. The new caption briefly defines the specific functions of Path-aware and Spatial Attention and clearly articulates their distinct roles within the overall architecture.
>
> **Paper Enhancement:** The caption of Figure 5 in paper of update version .
> > **Overview of Map Association Transformer (MAT)**. The framework processes vectorized roads in SD map and centerlines/boundaries in OP map through $N$ stacked layers containing Spatial Attention (for global context via curve-based serialization) and Path-Aware Attention (for topological alignment via path indexing). The Association Head then aggregates road features and calculates association probabilities with centerline tokens to generate the final navigation refinement result.

---

> ### Author Response · Authors · 2025-11-24
> **Comment for Review cbHh - 2**
>
> **Weaknesses 2**:
>
> Abbreviations are sometimes unclear or non-standard. For example, "Para." in Table 1 is ambiguous and should be explicitly defined (e.g., as "Parameters").
>
> **Response for Weaknesses 2**:
>
> Thank you for pointing this out. We have replaced ambiguous abbreviations (e.g., “Para.” → “Paradigm” in Table 1) with their full terms and conducted a full pass to ensure all abbreviations are standard and defined at first use.

---

> ### Author Response · Authors · 2025-11-24
> **Comment for Review cbHh - 3**
>
> **Weaknesses 3**:
>
> The section on "attention with vector serialization" is underdeveloped. The explanation is too brief, lacking the detail needed for the reader to fully understand the proposed mechanism.
>
> **Response for Weaknesses 3**:
>
> We thank the Reviewer for this critical observation. We acknowledge that the original explanation of "Attention with Vector Serialization" was indeed too concise. In the revised manuscript, we have significantly expanded this section to clarify both the fundamental motivation and the detailed implementation mechanism.
>
> 1. **Computational Constraints of Group Attention**: To meet the real-time inference demands of autonomous driving, MAT adopts a Group Attention mechanism. While this design reduces the computational complexity of standard global attention from $O(N^2)$ to linear $O(N)$, it introduces a hard constraint: the receptive field is strictly confined within local windows (or "groups"). Consequently, the permutation order of tokens directly dictates the composition of their "neighborhoods." If a random ordering were used, the attention mechanism would operate on fragmented, unrelated nodes, causing the model to fail in capturing meaningful contextual information.
>
> 2. **Vector Serialization as an Inductive Bias**: To maximize the effectiveness of feature interaction within this restricted receptive field, we introduce a structural prior. We conceptualize token sorting as a clustering strategy: it enforces that strongly correlated tokens are arranged adjacently in the sequence. When the sequence is sliced into groups, these correlated tokens naturally fall into the same attention bucket. This mechanism ensures that attention heads concentrate limited computational resources on processing high-value correlations rather than expending them on noise.
>
> 3. **Modeling Orthogonal Map Properties**: Road network data inherently possesses two distinct and mutually "orthogonal" attributes: geometry and topology. MAT explicitly models these core relationships by strictly implementing two specific ordering strategies:
>
> * **Spatial Order (based on Space-Filling Curves)**: Designed to encode geometric proximity: It ensures that physically adjacent entities, even if they are topologically distant or belong to different map layers (e.g., SD vs. OP maps), interact within the same group.
>
> * **Topological Order (based on Graph Traversal)**: Designed to encode connectivity. It ensures that predecessor and successor nodes interact within the same group.
>
> **Paper Enhancement**: To address this concern, we have refined the Sec. 5.2 and Sec 5.3 of Path-Aware Attention and Spatial Attention, incorporating a more detailed discussion on their design motivation and theoretical analysis.

---

> > ### Author Response · Authors · 2025-11-24
> > **Comment for Review cbHh - 4**
> >
> > **Question 1**:
> >
> > What’s the difference between MAT-T and MAT-L?
> >
> > **Response for Question 1**:
> >
> > We thank the Reviewer for the detailed observation regarding model configurations. To clarify, **MAT-T and MAT-L share an identical modular design** (incorporating Path-aware Attention, Spatial Attention, the boundary branch, and topological post-processing). The distinction lies solely in the configuration of Transformer layers, which offers different trade-offs between performance and latency.
> >
> > As detailed in **Table 16 of the Appendix**, both models maintain consistent structural parameters regarding channel dimensions, attention heads, MLP ratios, patch sizes, and spatial curve types. Specifically, they share:
> > * **Channels:** $[96, 192, 384, 768, 1536]$
> > * **Heads:** $[4, 4, 8, 8, 8]$
> > * **MLP Ratio:** $4$
> > * **Patch Size:** $1024$
> >
> > Furthermore, both models utilize identical loss functions (CE + CTC) and training hyperparameters (including learning rate, batch size, and data augmentation strategies).
> >
> > The **sole difference** is the number of Transformer blocks per stage:
> > * **MAT-T:** Blocks = $[2, 2, 2, 2, 2]$ (totaling **10** Transformer blocks). This is a lightweight version optimized for **real-time online navigation scenarios**.
> > * **MAT-L:** Blocks = $[4, 4, 4, 12, 4]$ (totaling **28** Transformer blocks). This represents a higher-capacity variant intended to demonstrate the **performance ceiling** of our method. Note that all ablation studies were conducted using MAT-L.
> >
> > **Paper Enhancement:**
> > Following your suggestion, we have added an explicit clarification in **Section 6 (Implementation Details)** to state that MAT-T and MAT-L share the exact same architecture and modules, differing only in block depth to address different deployment constraints (real-time vs. high-performance).

---

> > > ### Comment · Reviewer_cbHh · 2025-11-27
> > >
> > > Thanks for the author's revision and rebuttals. Many of my concerns are well addressed. I will accordingly raise my score.

---

> > > > ### Author Response · Authors · 2025-11-28
> > > >
> > > > Thank you very much for your prompt reply and positive feedback. We are glad that our revisions and responses have addressed your main concerns. We sincerely appreciate your valuable time and insightful comments, which have significantly improved the quality of our manuscript.

---

### Author Response · Authors · 2025-11-27
**Summary of Rebuttal and Response to Reviewers**

Dear PCs, SACs, ACs and Reviewers,

We thank the reviewers for their constructive feedback and the AC for handling our submission. Below, we provide a summary of the review status, the consensus on our contributions, and how we have addressed the specific concerns raised.

### 1. Summary of Scores and Rebuttal Status

The initial scores for our submission were **4, 6, and 8**. During the rebuttal phase, Reviewer cbHh (initially 4) responded to our comments and raised their score to **6**, acknowledging that our revisions addressed their concerns. The other two reviewers (Reviewer 9aX1 and Reviewer FhYL) have not yet provided further comments.

* **Reviewer cbHh:** **4 $\rightarrow$ 6**. The reviewer raised their score after acknowledging that our revisions (enhanced figures, clarified definitions, and theoretical expansions) well addressed their concerns.
* **Reviewer 9aX1:** **6**. The reviewer appreciated the "dataset-model-metric" trinity and solid experiments. Although they have not yet replied to our response, we have fully addressed their questions regarding robustness and generalization.
* **Reviewer FhYL:** **8**. The reviewer found our contribution "excellent" and the task important. Although they have not yet replied, we have provided the requested clarifications on definitions and ablation studies.

### 2. Consensus on Strengths

The reviewers reached a consensus on the value and novelty of our work:

* **Novel & Important Task  (Reviewers cbHh, 9aX1, FhYL):** Reviewers agreed that bridging SD Maps and Online Perception (OP) Maps is a critical and practical problem (ONR task) that addresses real-world needs in GIS and autonomous driving.
* **Systematic Contribution  (Reviewer 9aX1):** The "dataset-model-metric" trinity (OMA Dataset, MAT Model, NR P-R Metric) effectively fills the research gap in this field.
* **Novel and effectiveness method (Reviewers cbHh, FhYL):** The proposed MAT model with novel SA and PA module achieves low latency while maintaining high performance, and the experiments are solid and thorough.

### 3. Addressed Concerns and Solutions

We have rigorously addressed the questions raised by the reviewers in our revised manuscript and rebuttal:

**A. Robustness and Generalization (Reviewers 9aX1 & FhYL)**
* **Environmental Noise:** We conducted zero-shot experiments on **nuScenes-C** (Rain, Fog, Dark). Results show MAT maintains robust performance even under "Hard" degradation, with qualitative analysis demonstrating resilience to occlusion.
* **Cross-Generator Generalization:** We extended evaluations to include **MapTR** and **SeqGrowGraph**. MAT consistently outperforms baselines across different upstream generators.
* **SD Map Imperfections:** We added ablation studies simulating GPS errors (Global Shift) and map construction errors (Element Noise/Absence). The model showed strong resilience, with minimal performance drops even under significant noise.

**B. Metrics and Data Scalability (Reviewer 9aX1)**
* **Road Complexity:** We introduced a **Curvature-based Weighted Metric** to ensure fair evaluation across straight and complex curved roads.
* **Future dataset expansion:** We demonstrated a **"Human-in-the-Loop" pipeline**. Our data efficiency experiment showed that using only **5%** of training data yields comparable results to the full set.

**C. Methodological Clarifications & Related Work (Reviewers cbHh & FhYL)**
* **Visuals & Theory:** We improved **Figure 5** to better illustrate Path-aware and Spatial Attention and clarified the "Vector Serialization" mechanism (Group Attention & Inductive Bias).
* **Definitions:** We clarified the "Label Overlap" computation and the handling of ambiguous transition lines in ground truth generation.
* **Related Work:** We incorporated citations and discussions of related works (e.g., SMERF, TopoSD, P-MapNet) to better contextualize our approach regarding SD map utilization and noise resilience.

We believe the rebuttal has resolved the identified questions. We hope this summary assists the AC in the final decision-making process.

Best regards,

The Authors of Submission 6617

---

### Meta-Review · Area_Chair_ySJi · 2025-12-29

**Summary:**

This paper introduces Online Navigation Refinement, a new task designed to bridge the gap between Standard Definition maps and Online Perception maps. To support this, the authors present a dataset called Online Map Association (OMA), which is derived from nuScenes and OpenStreetMap. They provides the Map Association Transformer (MAT) model and a new metric named Navigation Refinement P-R to evaluate association accuracy. The contributions are sufficient, as it addresses a distinct bottleneck in autonomous driving by enabling low-cost, real-time lane guidance via SD-to-OP association. The proposed model demonstrates good performance, also achieving low latency while outperforming traditional map-matching baselines. All of three reviewers are quite positive and therefore, I recommend accepting this work.

**Reviewer Concerns:**

The authors successfully resolved the main issues raised by the reviewers during the rebuttal. They improved the clarity of the diagrams and model explanations that Reviewer `cbHh` initially found confusing, specifically fixing the descriptions of the attention mechanisms. To address Reviewer `9aX1`'s request for better testing, the authors added experiments showing the model works well in bad weather and with different map generation methods, while also adding a new metric for curved roads. They also satisfied Reviewer `FhYL` by explaining how the model handles errors in the standard map data and clarifying the definitions used for labeling, meaning there are no significant technical problems left.

**Reviewer Scores:**

Reviewer `cbHh` decided to increase their score from 4 to 6 because the authors fixed the unclear parts of the paper and better explained the attention mechanism. Reviewer `9aX1` maintained a positive score of 6, valuing the solid testing and the new evidence showing the system works on different data types. Reviewer `FhYL` gave a high score of 8 from the start and stayed very supportive, finding the new task important and the solution excellent after the authors provided the missing related work and definitions.

---

### Decision · Program_Chairs · 2026-01-26

Accept (Poster)